# LLMs on the Line: Data Determines Loss-to-Loss Scaling Laws

**Prasanna Mayilvahanan** [* 1 2 3 4]  **Thaddäus Wiedemer** [* 1 2 3 4]  **Sayak Mallick** [1 4]  **Matthias Bethge** [2 3 4]
**Wieland Brendel** [1 2 3]

## Abstract

Scaling laws guide the development of large language models (LLMs) by offering estimates for the optimal balance of model size, tokens, and compute. More recently, loss-to-loss scaling laws that relate losses across pretraining datasets and downstream tasks have emerged as a powerful tool for understanding and improving LLM performance and generalization. In this work, we investigate which factors most strongly influence loss-to-loss scaling. Our experiments reveal that the pretraining data determines the scaling trend. In contrast, model size, optimization hyperparameters, tokenizer and even significant architectural differences, such as between transformer-based models like Llama and state-space models like Mamba, generally have limited impact. Consequently, practitioners should carefully curate pretraining datasets for optimal downstream performance, while architectures and other settings can be freely optimized for training efficiency.

## 1. Introduction

Scaling laws have long guided Large Language Model (LLM) pretraining, determining model and data size under a fixed compute budget (Kaplan et al., 2020; Hoffmann et al., 2022; Grattafiori et al., 2024). Typically, scaling laws relate model performance, usually measured as training or validation loss, to total compute measured in floating point operations (FLOPs). FLOPs account for both parameter count and the number of training tokens. While useful for pretraining, scaling laws do not capture how well a model ultimately performs on downstream tasks (Gadre et al., 2024; Schaeffer et al., 2024; Du et al., 2025). Consequently, multiple works have begun to investigate *downstream scaling*

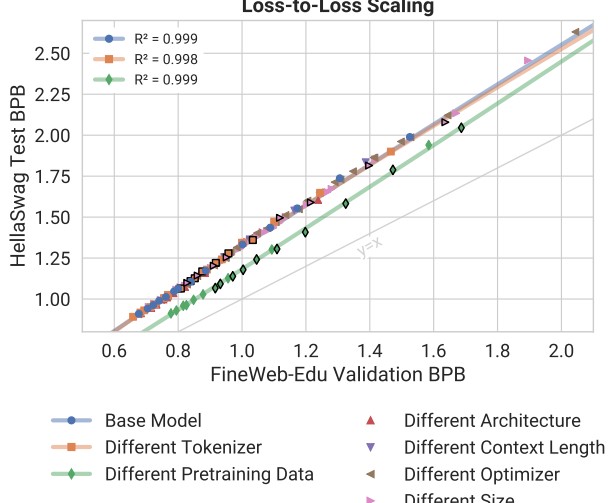

*Figure 1.* **LLMs' loss-to-loss scaling follows power laws primarily shaped by the choice of pretraining data**. Using Llama trained on FineWeb-Edu as a **baseline**, we intervene on various factors to assess their impact on train-to-test loss scaling. **Changing the pretraining data** has the largest effect. Changing the **tokenizer**, the architecture (e.g., from Llama to Mamba), model size, context length, and optimizer settings have little-to-no influence. For fair comparison across tokenizers, we normalize loss (negative-log-likelihood, nll) by bytes (bits-per-byte, BPB). We report goodness of fit for the fitted power laws as $R^2$. Power laws are fitted using all checkpoints, but only a random subset is shown to avoid visual clutter. Llama-7B checkpoints run for a few steps are highlighted with a black border.

*laws*: Scaling laws that directly predict downstream loss from FLOPs (Schaeffer et al., 2024; Gadre et al., 2024).

Brandfonbrener et al. (2024) show that *downstream scaling laws* can be decomposed into compute-to-train-loss scaling laws and (train)-loss-to-(test)-loss scaling laws. The combination of *compute-to-loss* and *loss-to-loss* scaling laws enables efficient and accurate prediction of a model's downstream performance. Moreover, holistic *downstream scaling laws* often optimize for a single task or average performance across tasks (Gadre et al., 2024; Schaeffer et al., 2024), whereas *loss-to-loss* (especially test-to-test) scaling laws can help tune a model's performance across a broader range of downstream tasks, e.g., to ensure broad or robust gen-

---

[*]Equal contribution [1]Max Planck Institute for Intelligent Systems [2]ELLIS Institute Tübingen [3]Tübingen AI Center [4]University of Tübingen. Correspondence to: Prasanna Mayilvahanan <prasanna.mayilvahanan@gmail.com>, Thaddäus Wiedemer <thaddaeus.wiedemer@gmail.com>.

*Proceedings of the $42^{nd}$ International Conference on Machine Learning*, Vancouver, Canada. PMLR 267, 2025. Copyright 2025 by the author(s).

eralization. Furthermore, train-to-test (or val-to-test) loss scaling laws characterize model generalization behavior. Specifically, these scaling laws are curves depicting the relationship between training or validation loss and downstream test performance, onto which we fit power-law functions.

While the impact of design choices like pretraining distribution, architecture, tokenizer, optimizer settings, etc. on compute-to-loss scaling laws is fairly well understood (Kaplan et al., 2020; Hoffmann et al., 2022; Tay et al., 2022; Wang et al., 2024; Porian et al., 2025; Du et al., 2025), a similar understanding is missing for loss-to-loss scaling laws. To close this gap, we extend the work of Brandfonbrener et al. (2024); Du et al. (2025), which analyze loss-to-loss relationships within a single architectural and training setup. Adding to that, our study systematically explores how multiple factors influence scaling laws across a diverse range of architectures and training configurations. Such an analysis on train-to-test (or val-to-test) scaling laws can help to understand factors that influence model generalization.

Our study draws inspiration from a body of work in robustness evaluation of vision (and later language) models (Taori et al., 2020; Miller et al., 2021; Fang et al., 2022; Awadalla et al., 2022). These works show that model performance on different distributions is frequently strongly correlated, and most model and training settings have little-to-no impact on the task-to-task scaling trend of model performance. We treat loss-to-loss curves similarly and perform a series of interventions using over 6000 model checkpoints to understand what design choices causally affect loss-to-loss scaling laws.

> We make three main observations, illustrated in Fig. 1:
>
> 1. LLMs' loss-to-loss scaling consistently follows shifted power laws.
>
> 2. Pretraining data is the most salient factor for these scaling laws.
>
> 3. In contrast, architecture and tokenizer generally play a minor role, while. model size, context length, and optimizer settings have little-to-no impact on loss-to-loss scaling.
>
> These observations imply that if two models with different training setups—but trained on the same data—achieve similar training losses, they will exhibit closely matched downstream test performance.

Our results indicate that common LLM architectures and training setups might encode very similar inductive biases, freeing practitioners to optimize them for training efficiency without adversely affecting downstream generalization.

## 2. From Scaling Laws to Interventions

**Compute-to-Train Scaling Laws**   Scaling laws aim to optimize model size and token allocation within a fixed compute budget (expressed in FLOPs) by modeling the relationship between parameters, training tokens, and training loss (Hestness et al., 2017; Kaplan et al., 2020; Hoffmann et al., 2022). However, these laws are inherently shaped by the data distribution, architecture, and optimization settings (Tay et al., 2022; Wang et al., 2024; Brandfonbrener et al., 2024; Porian et al., 2025), making their application across setups non-trivial.

**Compute-to-Downstream Scaling Laws**   Recent works extend scaling laws to directly predict downstream task performance from compute  (Gadre et al., 2024; Isik et al., 2024; Du et al., 2025). While some initial works attempt to map compute budgets to accuracy on individual tasks, multiple tasks, or aggregate benchmarks, this mapping is usually noisy due to several transformations in the accuracy computation that degrade the statistical relationship (Schaeffer et al., 2024). More recent efforts instead use the model's average loss on the correct answers of the task as a proxy (Madaan et al., 2024; Brandfonbrener et al., 2024). Such compute-to-downstream scaling laws provide a more practical perspective on scaling but are still specific to a given training setup.

**Loss-to-Loss Scaling Laws**   Loss-to-loss scaling laws aim to improve the transferability of scaling insights between training setups by examining the relationship between training (or validation) and test losses, between different validation losses, or between different test losses (Brandfonbrener et al., 2024). Typically, "training" or "validation" refers to general-purpose, open-text corpora used for pretraining, whereas "test" corresponds specifically to downstream tasks. This perspective is crucial for several reasons. First, train-to-train (or validation-to-validation) scaling implies how scaling laws transfer across datasets (Brandfonbrener et al., 2024). Second, incorporating train-to-test (or validation-to-test) scaling laws alongside compute-to-train scaling laws provides more precise insight into how compute budgets translate to downstream performance and can help study emergent abilities of models (Du et al., 2025). Third, while compute-to-loss scaling laws often target a single downstream task or average task performance, train-to-test and test-to-test scaling laws can help tune a model's performance across diverse tasks, e.g., to foster the development of generalist LLMs with a balanced task performance. Finally—and importantly—train-to-test (or validation-to-test) scaling laws serve as useful tools for understanding generalization.

**Accuracy on the Line**   Our work is inspired by robustness research in image classification. Prior studies (Taori et al., 2020; Miller et al., 2021; Fang et al., 2022) demonstrate a strong and consistent correlation between in-distribution and out-of-distribution (OOD) *accuracy* across various image classification models and settings. We are not the first to observe the similarity to LLMs, where recent works (Gadre et al., 2024; Brandfonbrener et al., 2024; Du et al., 2025) highlight strong scaling trends (linear or power-law-like) between *losses*. However, these studies are typically constrained to a single architecture or training setup. In contrast, we examine trends across a wide range of architectures and training conditions (see §4), showing for the first time that loss-to-loss scaling follows consistent laws across settings.

**Robustness Interventions**   Accuracy-to-accuracy relationships in the vision, vision-language, and language domains have also been used to study how scaling laws shift under robustness interventions like dataset size, adversarial training, architectural details, loss functions, supervision type, or OOD shifts (Taori et al., 2020; Fang et al., 2022; Awadalla et al., 2022; Mayilvahanan et al., 2024a;b; Wiedemer et al., 2024). For vision-language models, Taori et al. (2020); Fang et al. (2022) find that most interventions do not impact OOD performance; only increasing data diversity has a significant positive impact. Their findings suggest that curating better datasets is crucial for training vision and vision-language models that generalize broadly.

Motivated by these insights, we aim to uncover the factors determining loss-to-loss scaling to better understand what matters for generalization. Our insights complement the findings from Awadalla et al. (2022), who show that accuracy-accuracy scaling trends in comprehension tasks are agnostic to architecture type (e.g., encoder-only, encoder-decoder, decoder-only) after fine-tuning. In contrast to their study, we focus on zero-shot generalization across a diverse set of tasks, specifically investigating state-of-the-art decoder-only architectures such as GPT (Radford et al., 2019), Llama (Grattafiori et al., 2024), and Mamba (Gu & Dao, 2024; Dao & Gu, 2024).

## 3. Fitting Loss-to-Loss Scaling Laws

We focus our analysis on train-to-train and train-to-test scaling. Combined with known compute-to-train scaling laws, these loss-to-loss scaling laws paint a complete picture of a model's downstream performance given a compute budget and characterize a model's downstream performance distribution across tasks (Brandfonbrener et al., 2024).

As is standard in the recent literature, we report test loss as a proxy for downstream performance. Following Brandfonbrener et al. (2024); Madaan et al. (2024); Schaeffer et al. (2024), we track the test loss as a model's loss on only the

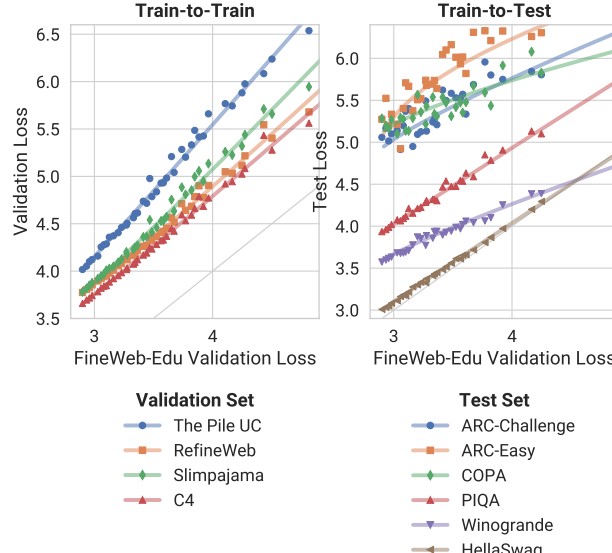

**Validation Set**
- The Pile UC
- RefineWeb
- Slimpajama
- C4

**Test Set**
- ARC-Challenge
- ARC-Easy
- COPA
- PIQA
- Winogrande
- HellaSwag

*Figure 2.* **Loss-to-loss scaling consistently obeys power laws.** We extend results from Brandfonbrener et al. (2024) to many architectures, training settings, and validation/test sets. We show illustrative shifted power laws for Mamba trained on FineWeb-Edu here; more configurations and test sets can be found in App. E. For clarity, scatter plots display a random sample of all data points; all points are used to fit the scaling laws.

correct answer given the question as context. This is sometimes called the *cloze formulation* of a task since the model is essentially evaluated on its ability to fill in blanks.

Brandfonbrener et al. (2024) predict train-to-train and train-to-test scaling laws to follow a shifted power law[1]

$$L_y\left(f_p^{N,D}\right) \approx K \cdot \left(L_x\left(f_p^{N,D}\right) - E_{x|p}\right)^{\kappa} + E_{y|p}, \quad (1)$$

where $L_x, L_y$ are the losses on datasets $\mathcal{D}_x, \mathcal{D}_y$ shown on the x- and y-axis. $f_p^{N,D}$ is a model trained with $N$ parameters on $D$ tokens on the pretraining set $\mathcal{D}_p$, and $K$ and $\kappa$ are parameters to be fit. $E_{x|p}, E_{y|p}$ are the irreducible errors (i.e., minimum loss) that $f_p$ trained on $\mathcal{D}_p$ can achieve on the datasets $\mathcal{D}_x, \mathcal{D}_y$.

Given a model configuration, we collect 200 to 800 checkpoints throughout training, across model sizes, and for various seeds. All points are used to first estimate $E_{x|p}, E_{y|p}$ from individual compute-to-loss scaling laws and then fit $K, \kappa$. Refer to App. A and App. B for more details.

Note that to study the impact of various training settings, the x- and y-axis show losses on *different datasets* of *the same model*. This should not be confused with some figures in Brandfonbrener et al. (2024) where the x- and y-axis show losses of *different compute-matched models*.

---

[1]Eq. (1) here follows from Brandfonbrener et al. (2024) Eq. (4) when assuming an irreducible error as in Eqs. (6, 7); see App. A.

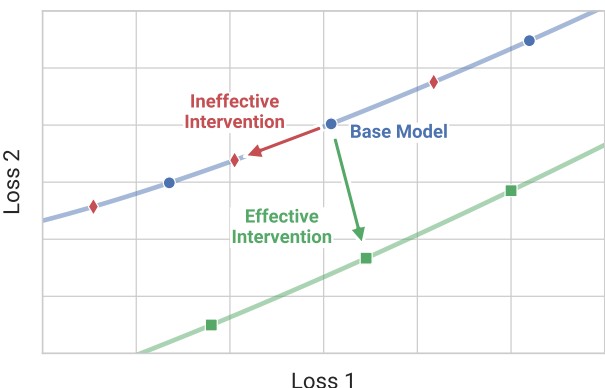

*Figure 3.* **Schematic of our causal analysis**. Checkpoints of a **base model** trained on different numbers of tokens and with different seeds lie on the same loss-to-loss line. Better-performing models (typically with higher compute) achieve lower loss (towards the bottom left). We intervene on training settings (e.g., pretraining data, architecture, etc.) and retrain from scratch, yielding new models that again constitute loss-to-loss lines. An **effective intervention** produces models on a new line; an **ineffective intervention** yields models that lie on the line of the **base model**.

With this setup, we can now analyze the loss-to-loss scaling laws of models trained with different configurations. Brandfonbrener et al. (2024) only showed loss-to-loss scaling a single architecture with a fixed training recipe: Olmo (Groeneveld et al., 2024). We extend their analysis by multiple architectures, pretraining sets, tokenizers, and training settings, all listed in §4. As an illustrative example, we show loss-to-loss scaling for Mamba trained on FineWeb-Edu in Fig. 2; more examples with additional test sets are listed in App. E and throughout §4.

Overall, across models, datasets, tokenizers, and optimization hyperparameters shifted power laws describe loss-to-loss scaling well (Eq. (1)). On some datasets, the performance of models with high loss (towards the top right of each curve) is not captured perfectly by the power law formulation proposed by Brandfonbrener et al. (2024). This is unsurprising, given that these data points typically represent models in early training stages but might hint at a refined formulation of Eq. (1) for the high-loss regime.

Note also that loss-to-loss scaling follows a power law even for datasets on which the model never reaches high accuracy. E.g., Mamba in Fig. 2 never surpasses chance performance on ARC-Challenge, yet Eq. (1) describes the test loss equally well. This underlines the usefulness of loss-to-loss scaling laws to study model behavior.

> **Takeaway 1** Across architectures and training settings, loss-to-loss scaling generally follows a shifted power law as described in Eq. (1) and illustrated in Fig. 2.

## 4. A Causal Analysis of Loss-to-Loss Scaling

We now perform interventions on the model and training configurations to find what factors cause the exact shape of loss-to-loss scaling laws.

Our basic procedure is outlined in Fig. 3. As mentioned in §2, our approach is motivated by similar studies in the robustness literature. In contrast to that setting, here we lack paired in-distribution and out-of-distribution datasets. Instead, we simply consider all combinations of validation and test sets. Typically, "validation" refers to general-purpose open-text corpora used during pretraining, whereas "test" corresponds specifically to downstream tasks. For ease of visualization when intervening on the pretraining data, we always show FineWeb-Edu validation loss on the x-axis, even for models trained on different pretraining distributions. This choice is arbitrary and does not affect our results; see App. F. Similarly, we here report results for scaling laws of *average* validation and test loss; results for individual losses can be found in App. G.

For our analysis, we consider the impact of pretraining data, tokenizer, architecture, model size, context length, and optimizer settings.

**Pretraining Sets** Our models are trained on FineWeb-Edu (Penedo et al., 2024), C4 (Dodge et al., 2021), and an uncopyrighted version of The Pile dubbed The Pile UC. Some models from Hugging Face are trained on the original version of The Pile (Gao et al., 2020) and The Pile Deduped (Biderman et al., 2023), a deduplicated version.

**Validation Sets** Models are evaluated on 5000 sequences sampled from the validation sets of FineWeb-Edu, C4, The Pile, RefinedWeb (Penedo et al., 2023), and SlimPajama (Shen et al., 2024).

**Test Sets** We use LM Evaluation Harness framework (Gao et al., 2024) to assess model performance on HellaSwag (Zellers et al., 2019), COPA (Gordon et al., 2011), WinoGrande (Sakaguchi et al., 2019), PIQA (Bisk et al., 2019), SocialIQA (Sap et al., 2019), CommonsenseQA (Talmor et al., 2019), MMLU (Hendrycks et al., 2021), as well as ARC-Easy and ARC-Challenge (Clark et al., 2018).

**Models** We train Llama-3 (Grattafiori et al., 2024) with $417\,\text{M}$ parameters and Mamba (Gu & Dao, 2024) with $420\,\text{M}$ parameters using the Lingua framework (Videau et al., 2024), following Chinchilla scaling laws (Hoffmann et al., 2022). We supplement our analysis with pretrained GPT (Black et al., 2021; 2022; Biderman et al., 2023), Llama (Penedo et al., 2024), and Mamba (Gu & Dao, 2024; Dao & Gu, 2024) variants from Hugging Face (Wolf et al., 2020). We present more details on the models in App. C.

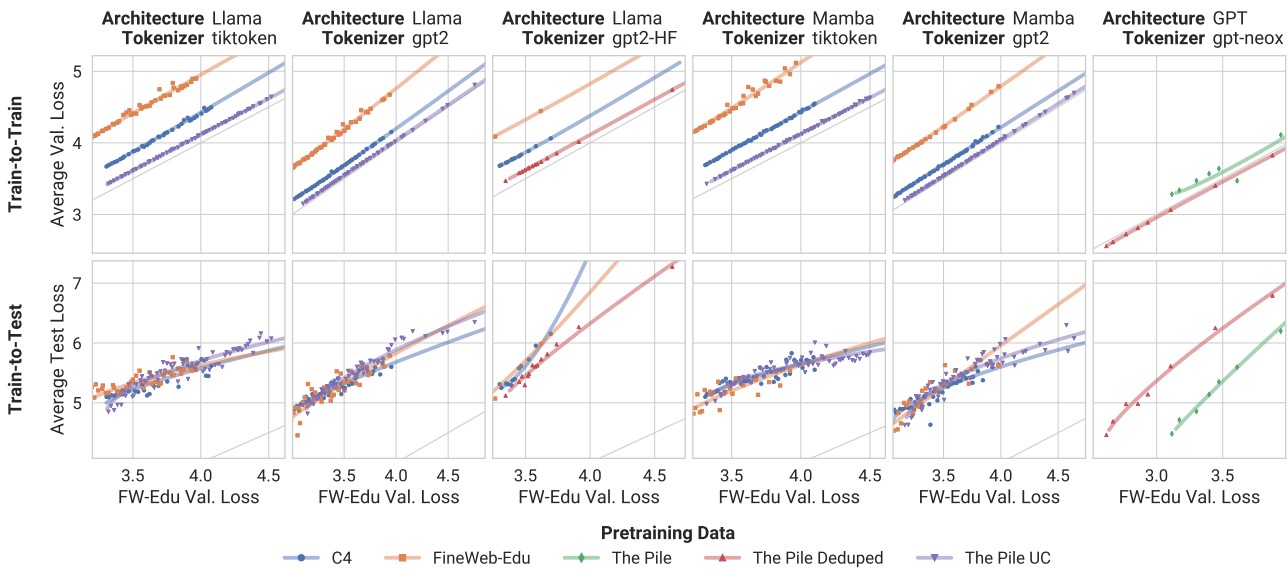

*Figure 4.* **Pretraining data has a substantial impact on loss-to-loss scaling laws**. Models are matched on architecture and tokenizer.

**Tokenizers**   We train Llama and Mamba with either a `tiktoken` tokenizer (128 k vocabulary size) or the `gpt2` tokenizer (50 257 vocabulary size). Pretrained models from Hugging Face use an almost identical GPT-2 tokenizer, dubbed `gpt2-HF`. This version does not explicitly pad text with beginning and end-of-sequence tokens. A few Hugging Face GPT models instead use the `gpt-neox` tokenizer with a slightly different vocab size of 50 254, which results in a different internal mapping compared to `gpt2`,

### 4.1. Pretraining Data, Tokenizer, and Architecture

First, we jointly examine the effect of pretraining data, architecture, and tokenizer. Since we face limited compute to train models from scratch, we do not have checkpoints for all possible combinations of these factors. Instead, we analyze the effect of an intervention on each factor when matching models in the two other factors. Note that we do not have sufficient checkpoints for some Hugging Face models to fit a power law. Nevertheless, in all these cases, the available data points follow a clearly discernible trend.

**Effect of Pretraining Data**   Fig. 4 illustrates the substantial impact pretraining data has on loss-to-loss scaling. Across architectures and compute (in different columns), changing the pretraining data leads to a large shift in the loss-to-loss curve. Notably, in the last column, we compare Hugging Face models trained on The Pile versus its deduplicated variant. Models trained on these two datasets fall onto slightly different scaling curves, indicating that deduplication has meaningfully altered the underlying distribution and thereby also affecting the loss-to-loss scaling.

> **Takeaway 2** With fixed architecture and tokenizer, changing the pretraining data leads to *substantial* shifts in loss-to-loss scaling laws; see Fig. 4.

**Effect of Tokenizer**   Fig. 5 illustrates the impact of tokenizer choice on loss-to-loss scaling laws. To enable fair comparisons between models trained with different tokenizers, we measure performance using bits-per-byte (BPB) Gao et al. (2020). In our experiments, we observe only minor tokenizer-induced variations in scaling curves, as they remain closely aligned. However, the pretrained 'ablation models' from Penedo et al. (2024) (first and second columns) notably deviate from our curves. One key distinction is that the models from Penedo et al. (2024) utilize 'weight tying' between the token embeddings in the first layer and the output token predictions in the final layer, whereas our models do not. This difference in training setup could explain the observed discrepancy in the scaling curves. We leave a detailed investigation of this hypothesis for future work.

> **Takeaway 3** With fixed architecture and pretraining data, changing the tokenizer generally leads to *minor* changes in loss-to-loss scaling laws; see Fig. 5.

**Effect of Architecture**   Lastly, Fig. 6 illustrates that changing the architecture results in only very slight changes in the loss-to-loss curves across pretraining data and tokenizer settings. Unlike pretraining data and tokenizer, architecture has little influence on train-to-train and train-to-test scaling. This is particularly surprising given the significant architectural differences between Llama or GPT (transformer-based

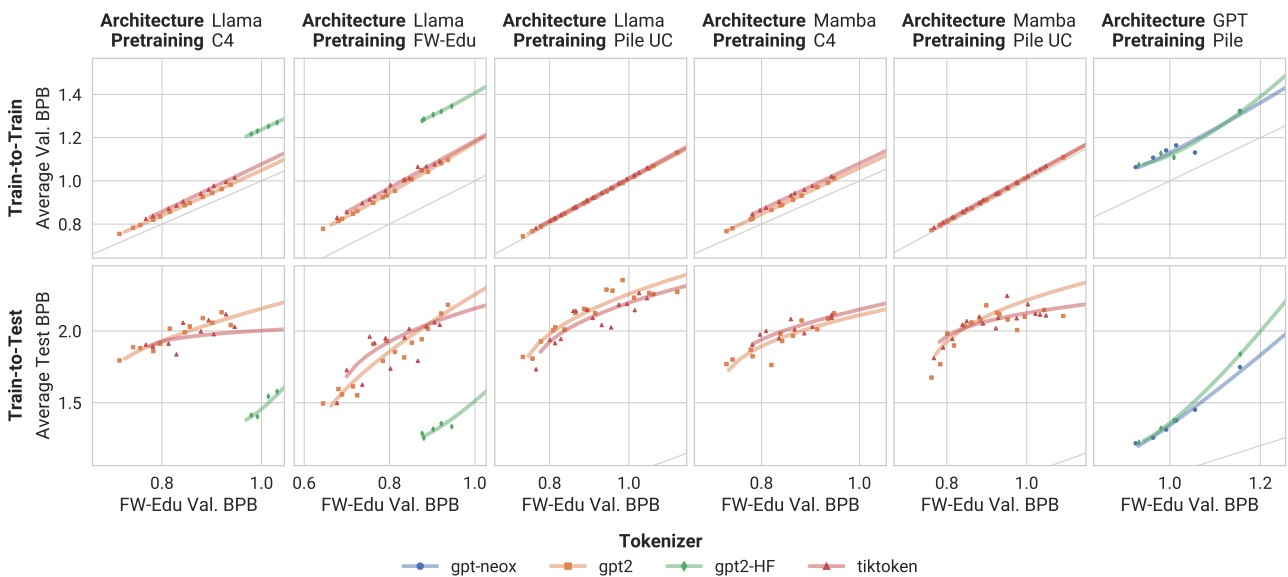

*Figure 5.* **The tokenizer has a minor impact on loss-to-loss scaling laws**. Models are matched on pretraining data and architecture.

models) and Mamba (a state-space model). These results raise an important question: Do current architectures encode distinct inductive biases or converge to similar solutions given the same training data? Further research is needed to understand the implications of this finding.

> **Takeaway 4** With fixed pretraining data and tokenizer, changing the architecture has *limited* impact on loss-to-loss scaling laws — raising questions about the distinctiveness of their inductive biases; see Fig. 6.

### 4.2. Model Size, Context Length, and Optimization

We now examine the effect of other common design decisions, such as the number or width of layers, the context length, optimizer, learning schedule, learning rate, and weight decay. In contrast to §4.1, we can perform these interventions separately since we can compare among our own Llama and Mamba models whose training settings are matched by default.

To provide a more succinct overview, we only show train-to-train scaling laws in this section; additional train-to-test scaling laws for the same intervention can be found in App. H. We also do not show fitted power laws here since we display many more models per plot than in §4.1, and the scaling trends are clearly discernible.

**Effect of Model Size** We first examine the influence of model size by training Llama and Mamba models with varying depths and widths (see App. C for details). Fig. 7 shows the results: Despite significant differences in parameter count, the loss-to-loss scaling trends remain unchanged.

These findings align well with Du et al. (2025), who observed that model size has little effect on loss-to-loss scaling for GPT models. We extend this conclusion to Llama and Mamba and across multiple pretraining distributions.

**Effect of Context Length** We next investigate the effect of varying the context length between 1024, 2048, and 3076 tokens. As shown in Fig. 8, this change does not meaningfully affect the loss-to-loss scaling curves.

**Effect of Optimization Settings** Finally, we evaluate a range of common optimization settings: We consider the Adam (Kingma & Ba, 2017) and AdamW (Loshchilov & Hutter, 2019) optimizers, cosine (Loshchilov & Hutter, 2017) and WSD (Hu et al., 2024) schedules, learning rates of $3e-4$ and $3e-3$, and a weight decay of $0.1$ or $3.3e-2$. In our training setup, models using the Adam optimizer generally did not converge, and we exclude them from the analysis. Variations of the other settings do not affect loss-to-loss scaling coefficients, as shown in Fig. 9.

> **Takeaway 5** Model size (Fig. 7), context length (Fig. 8), and optimization settings (Fig. 9) have *negligible* impact on loss-to-loss scaling laws.

Given the limited impact of the factors studied in this section, the conclusions from §4.1 should generalize well across variations in model size, context length, and optimization settings. For example, the substantial impact of the pretraining distribution can also be observed in Figs. 7 and 8.

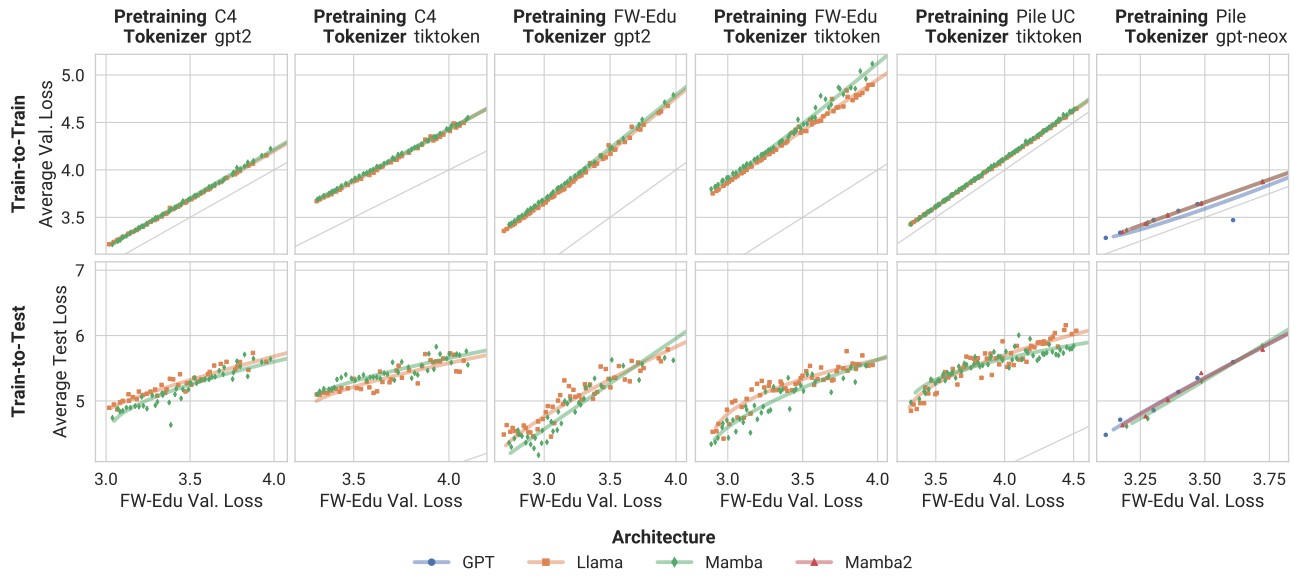

*Figure 6.* **Architecture has limited impact on loss-to-loss scaling laws**. Models are matched on pretraining data and tokenizer.

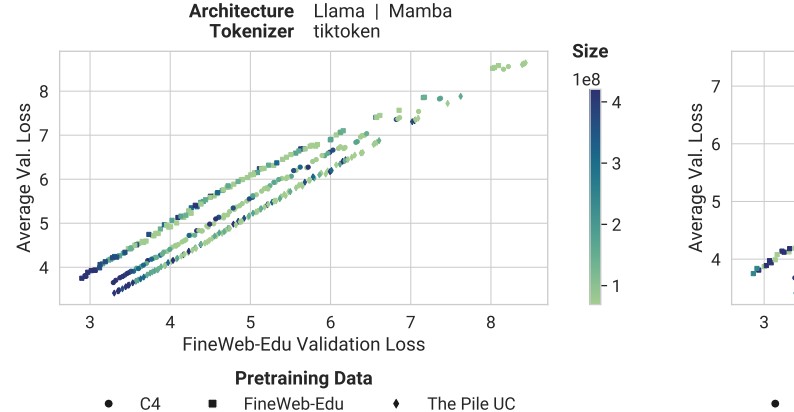

*Figure 7.* **Model size does not affect loss-to-loss scaling**. The distinct lines correspond to different pretraining distributions (see Fig. 4), reinforcing that their influence is consistent across scales.

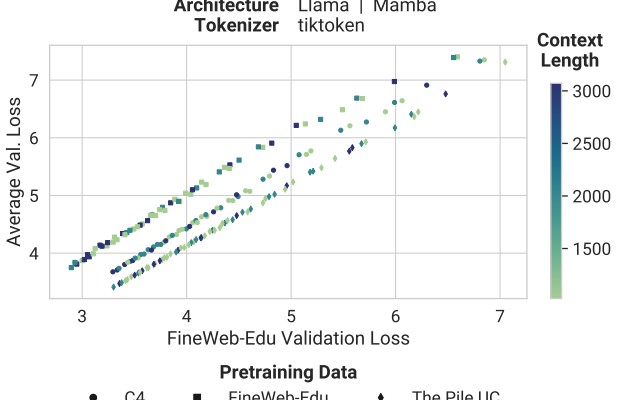

*Figure 8.* **Context length does not affect loss-to-loss scaling**. Again, distinct lines correspond to different pretraining distributions (compare Fig. 4), validating their consistent impact.

## 5. Point-wise Comparisons

In our experiments so far, we have demonstrated that nearly all factors we investigated—except the choice of pretraining data—have minimal to no effect on loss-to-loss scaling. This implies a useful generalization: if two distinct training setups achieve similar training or validation losses on the same pretraining data, they will exhibit similar test losses across many downstream tasks. To explicitly illustrate this point, we present detailed point-wise comparisons between Mamba and Llama models trained on identical datasets in Table 1, highlighting their closely matched downstream performance.

## 6. Discussion and Future Work

Our findings add to the understanding of loss-to-loss scaling laws and reinforce prior results from vision and vision-language research (Taori et al., 2020; Fang et al., 2022) on the importance of choosing the pretraining data.

**Implications for Optimizing Downstream Performance**
Our results emphasize that the data distribution is the key for achieving a desireable loss-to-loss scaling and a in turn achieve a great downstream performance. Conversely, since architecture has little impact on the train-to-test conversion, it can be freely optimized for better compute scaling without affecting downstream scaling or performance.

*Table 1.* Comparison of Mamba and Llama models. Models are compared after seeing approximately 8 Billion Tokens of their pretraining data, at two different model sizes. We observe similar downstream validation and test losses. PT: Pretraining, SPJ: SlimPajama, RW: RefinedWeb, FW-E: FineWeb-Edu, ARC-C: ARC-Challenge, and ARC-E: ARC-Easy.

| Model Type | Size | PT. Data | Validation Losses | | | | | Test Task Losses | | | | | Avg Val. Loss | Avg Test Task Loss |
|---|---|---|---|---|---|---|---|---|---|---|---|---|---|---|
| | | | C4 | Pile UC | FW-E | RW | SPJ | ARC-C | ARC-E | HellaSwag | MMLU | PIQA | | |
| Mamba | 421M | FW-Edu | 3.66 | 4.02 | 2.90 | 3.77 | 3.78 | 5.06 | 5.27 | 3.01 | 2.14 | 3.94 | 3.63 | 3.88 |
| Llama | 417M | FW-Edu | 3.66 | 3.89 | 2.90 | 3.74 | 3.72 | 5.26 | 5.73 | 3.02 | 2.24 | 3.96 | 3.58 | 4.04 |
| Mamba | 421M | Pile UC | 3.70 | 2.75 | 3.32 | 3.76 | 3.50 | 5.58 | 6.07 | 3.23 | 4.51 | 4.14 | 3.41 | 4.70 |
| Llama | 417M | Pile UC | 3.71 | 2.75 | 3.32 | 3.77 | 3.49 | 5.55 | 5.96 | 3.26 | 4.46 | 4.18 | 3.41 | 4.68 |
| Mamba | 421M | C4 | 3.17 | 4.13 | 3.31 | 3.85 | 3.61 | 5.69 | 6.34 | 2.94 | 5.04 | 3.73 | 3.61 | 4.75 |
| Llama | 417M | C4 | 3.17 | 4.05 | 3.29 | 3.82 | 3.57 | 5.63 | 6.25 | 2.95 | 5.33 | 3.76 | 3.58 | 4.78 |
| Mamba | 368M | FW-Edu | 3.74 | 4.15 | 2.98 | 3.85 | 3.86 | 5.13 | 5.42 | 3.10 | 3.26 | 4.01 | 3.72 | 4.18 |
| Llama | 365M | FW-Edu | 3.74 | 3.98 | 2.98 | 3.84 | 3.81 | 5.39 | 5.78 | 3.10 | 3.52 | 4.04 | 3.67 | 4.36 |
| Mamba | 368M | Pile UC | 3.78 | 2.83 | 3.40 | 3.85 | 3.59 | 5.75 | 6.35 | 3.31 | 5.08 | 4.17 | 3.49 | 4.93 |
| Llama | 365M | Pile UC | 3.79 | 2.82 | 3.41 | 3.86 | 3.58 | 5.97 | 6.56 | 3.32 | 4.96 | 4.28 | 3.49 | 5.02 |
| Mamba | 368M | C4 | 3.25 | 4.23 | 3.39 | 3.92 | 3.69 | 5.92 | 6.66 | 3.03 | 5.45 | 3.79 | 3.70 | 4.97 |
| Llama | 365M | C4 | 3.27 | 4.13 | 3.38 | 3.91 | 3.65 | 5.77 | 6.33 | 3.05 | 5.56 | 3.90 | 3.67 | 4.92 |

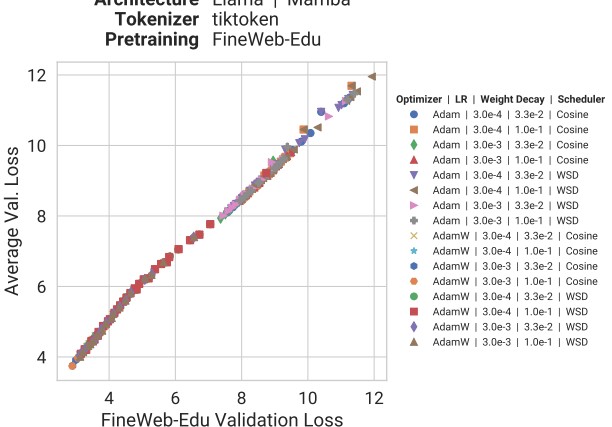

*Figure 9.* **Optimization settings do not affect loss-to-loss scaling**.

**Implications for Balancing Performance** If the aim is not only optimal average downstream performance but also a specific weighting between different tasks, e.g., to ensure a balanced downstream performance, individual train-to-test scaling laws can be used to tune a model's performance. Here, too, the pretraining data has the largest impact and practitioners should thus consider the final application of their model already during the data curation stage. Ultimately, our findings underscore that pretraining data curation, rather than architectural innovation, can be the primary driver in developing robust, generalist models.

**On Architectural Biases** The limited impact of even drastically different architectures on loss-to-loss scaling behavior illustrated in §4.1 and Fig. 6 suggest that architectures trained on the same data may implicitly learn highly similar representations. This might seem intuitive, as all models minimize the same loss function. One might expect them to converge toward comparable solutions when the training loss approaches zero (Roeder et al., 2020). However, even checkpoints of our smaller models, when trained on fewer tokens, follow the same scaling across architectures. Understanding whether this implies representational and behavioral similarity remains an intriguing open question. Beyond this, it remains to be seen whether it is possible to formulate architectures that fit the data well but exhibit different scaling trends.

**On Other Training Paradigms** Our study intentionally focuses on models trained with standard loss functions and conventional training settings to guide practitioners. The limited impact of existing paradigms does not preclude innovative training approaches from improving loss-to-loss scaling. In fact, a recent work by Saunshi et al. (2024) demonstrates that gradually increasing model depth and initializing based on layers from a smaller model produces markedly different scaling behavior, particularly in how perplexity translates to downstream accuracy. Similar structured growth approaches could offer new pathways for improving scaling efficiency and generalization for decoder-only LLMs trained with next-token prediction. Additionally, in §4.1 we observed that weight tying could play a potential role in shifting these generalization curves. We leave these analyses for future work.

**On the Exhaustiveness of Interventions in §4.1** Our study clearly distinguishes between factors with substantial and limited impact on loss-to-loss scaling. While our conclusions are inherently shaped by the specific settings we explored, the observed trends provide strong empirical evidence for these distinctions. Given the strong and consistent impact of pretraining data and tokenizer, we can confidently conclude that these interventions affect loss-to-loss scaling. While we observed only a limited impact of the architecture,

this effect was also consistent across major state-of-the-art architectures including Llama, GPT, and Mamba — which collectively represent the dominant paradigms in large-scale language modeling. Given this exhaustive set, it is hard to argue that other architectures would meaningfully alter loss-to-loss scaling.

**On the Exhaustiveness of Interventions in §4.2** Across the wide range of size configurations (App. C) we test, all models exhibit very consistent loss-to-loss scaling. Similarly, the effect we observed for different context lengths is very consistent within our test range (1024, 2048, 3076), which aligns with commonly used configurations (Black et al., 2021; Wang & Komatsuzaki, 2021; Biderman et al., 2023; Penedo et al., 2024; Black et al., 2022). While we acknowledge the possibility that larger models or longer context lengths could influence loss-to-loss scaling, such an effect — if present — is unlikely. For optimization settings, we again consider configurations widely used in LLM training (Shoeybi et al., 2020; Karpathy, 2022; Videau et al., 2024), including variations in optimizer type, learning rate, weight decay, and scheduling. While our results indicate that these choices do not meaningfully alter loss-to-loss scaling within the explored settings, we acknowledge that the space of optimization techniques is vast, and our list is not exhaustive. It remains possible that a principled optimization strategy, different from current best practices, could induce new scaling behaviors. However, our findings suggest that optimization settings are not a primary driver of loss-to-loss scaling trends within the bounds of conventional language model training.

**On the Goodness of Fit of Scaling Laws** For a given model size, we train on a number of tokens up to the Chinchilla-optimal amount (e.g., $8.4\,\mathrm{B}$ tokens for a $420\,\mathrm{M}$ parameter model) and maintain a constant warmup of $5000$ steps, a learning rate of $3\mathrm{e}{-3}$, and a one-cycle cosine decay schedule. We use intermediate checkpoints as a proxy for models trained on fewer tokens. While (Hoffmann et al., 2022) suggest adapting the scheduler in this case to align with the number of tokens, we are constrained by compute and cannot train thousands of models from scratch. While using intermediate checkpoints alongside models of different sizes may influence the specific shape of the fitted compute-to-loss scaling laws, we find that the overall quality of fit benefits greatly from the additional data points. Since we only use compute-to-loss scaling laws to estimate the entropy terms in Eq. (1) and are interested primarily in the impact of interventions on loss-to-loss scaling, we do not expect this choice to significantly impact our conclusions. We also not that using intermediate checkpoints for fitting scaling laws is not unprecedented in the literature (Schaeffer et al., 2024; Brandfonbrener et al., 2024).

## 7. Conclusion

In this work, we systematically investigate loss-to-loss scaling in LLMs, identifying key factors that shape its behavior. Our large-scale interventional analysis — spanning over 6000 model checkpoints across architectures, tokenizers, and training setups — reveals that loss-to-loss scaling consistently follows shifted power-law trends, enabling predicting test performance from training loss.

We identify pretraining data and tokenizer as the dominant factors shaping these scaling laws, highlighting the importance of data curation. Architecture has limited impact, with models as different as LLaMA (transformer-based) and Mamba (a state-space model) exhibiting nearly identical scaling when trained on the same data and tokenizer. Model size, context length, and optimization settings have negligible influence, such that loss-to-loss scaling remains stable across different configurations.

Our findings underline the importance of pretraining data for downstream performance and robustness and suggest that different LLM might share similar architectural biases. Given our observations, practitioners should prioritize curating high-quality pretraining data to optimize downstream performance, while architectures and training settings can be adjusted freely for efficiency.

## Impact Statement

This paper presents work whose goal is to advance the field of Machine Learning. There are many potential societal consequences of our work, none of which we feel must be specifically highlighted here.

## Author Contributions

This project was co-led and coordinated by PM and TW. TW and PM jointly developed the method, incorporating insights from WB and MB. PM was responsible for training and evaluating the models. SM assisted with certain evaluations and provided some details for Hugging Face models. TW analyzed the data and conducted the scaling law and loss-to-loss fits. Both TW and PM contributed to writing the manuscript, with feedback from WB. TW created all the figures with feedback from PM.

## Acknowledgements

We would like to thank (in alphabetical order) Ameya Prabhu, Attila Juhos, Evgenia Rusak, Fanfei Li, Jack Brady, Thomas Klein, and Vishaal Udandarao for helpful discussions and feedback.

This work was supported by the German Federal Ministry of Education and Research (BMBF): Tübingen AI Center, FKZ: 01IS18039A. WB acknowledges financial support via an Emmy Noether Grant funded by the German Research Foundation (DFG) under grant no. BR 6382/1-1 and via the Open Philantropy Foundation funded by the Good Ventures Foundation. WB is a member of the Machine Learning Cluster of Excellence, EXC number 2064/1 – Project number 390727645. This research utilized compute resources at the Tübingen Machine Learning Cloud, DFG FKZ INST 37/1057-1 FUGG.

We thank the International Max Planck Research School for Intelligent Systems (IMPRS-IS) for supporting PM and TW.

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

## A. Scaling Law Details

We adopt the compute-to-loss scaling law formulation from Brandfonbrener et al. (2024) Eq. (4):

$$L\left(f^{(N,D)}\right) = E + \left(\left(\frac{A}{N}\right)^{\frac{\alpha}{\beta}} + \frac{B}{D}\right)^{\beta}, \tag{2}$$

where $f^{(N,D)}$ is a model with $N$ parameters trained on $D$ tokens and $E, A, B, \alpha, \beta$ are parameters to be fit. Notably, the irreducible error $E$ captures the minimum loss possible for model $f$ in the limit of infinite model and data size.

By default, Eq. (2) is fit using the training or validation loss. However, as demonstrated by Brandfonbrener et al. (2024) and our experiments, we can alternatively predict the loss $L_x$ on dataset $\mathcal{D}_x$ achieved by model $f_p^{(N,D)}$ trained on the pretraining set $\mathcal{D}_p$:

$$L_x\left(f_p^{(N,D)}\right) = E_{x|p} + \left(\left(\frac{A}{N}\right)^{\frac{\alpha}{\beta}} + \frac{B}{D}\right)^{\beta}. \tag{3}$$

As in Brandfonbrener et al. (2024) Eq. (7), the irreducible error

$$E_{x|p} = L_x(f_p^*) \tag{4}$$

then captures the minimum possible loss on $\mathcal{D}_x$ of a model trained on $\mathcal{D}_p$.

With that, we can formulate the loss-to-loss scaling law for arbitrary combinations of pretraining data and two test or validation sets, as stated in Eq. (1).

## B. Fitting Details

As explained in §3, we fit loss-to-loss scaling law in Eq. (1) by collecting 200 to 800 model checkpoints throughout training and across model sizes and seeds.

For each line in a plot corresponding to a loss-to-loss scaling law from Eq. (1), we first fit the two compute-to-loss scaling laws $L_x(f_p^{(N,D)})$ and $L_y(f_p^{(N,D)})$ given by Eq. (3). This yields estimates for the irreducible errors $E_{x|p}, E_{y|p}$, which correspond to the minimum x- and y-value of the loss-to-loss line. We use SciPy's default `curve_fit` optimizer for fitting (Virtanen et al., 2020). In rare cases when all checkpoints have the same number of parameters $N$ or same number of tokens $D$ (this is the case only for a small subset of the Hugging Facemodels) and a compute-to-loss scaling law cannot be fitted, we instead estimate the irreducible error as the minimum loss achieved:

$$E_{x|p} = \min_{N,D} L_x\left(f_p^{(N,D)}\right). \tag{5}$$

We show example compute-to-loss fits for some of the loss-to-loss scaling laws from Fig. 2 in Fig. 10.

With $E_{x|p}, E_{y|p}$ from the compute-to-loss fits, we again use SciPy's `curve_fit` to fit $K, \kappa$ for the loss-to-loss scaling law from Eq. (1).

## C. Model Details

Table 2 supplements the discussion of the models used for our study from §4.1 with additional details.

## D. Quantitative Analysis of Interventions

We quantify the impact of different interventions as the area between fitted curves in Table 4. Pretraining data clearly has the biggest impact on the scaling laws.

## E. Loss-to-Loss Scaling Across Settings

We supplement Fig. 2 from §3 with additional architecture-pretraining pairings in Figs. 11 to 16.

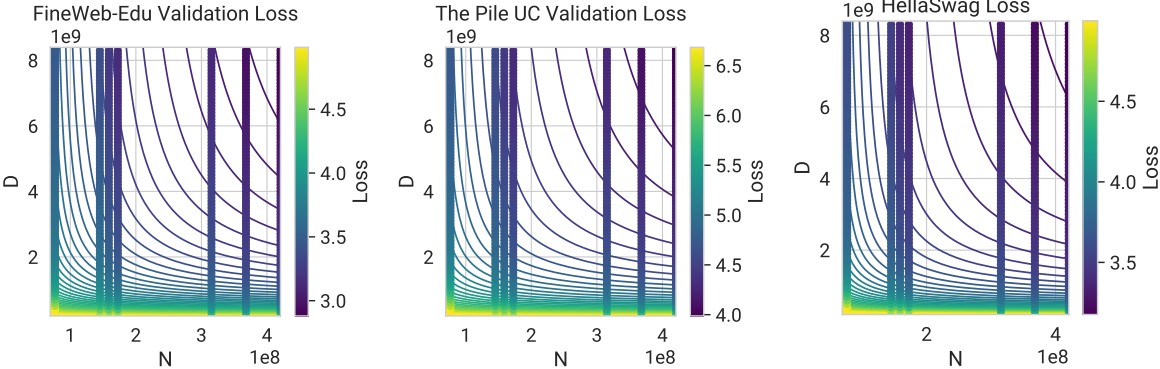

*Figure 10.* **Example compute-to-loss scaling law fits**. Each loss-to-loss scaling law requires fitting two compute-to-loss scaling laws to estimate $E_{x|p}, E_{y|p}$. The three fits here are used for the The Pile UC and HellaSwag curves in Fig. 2, which showed curves for FineWeb-Edu-trained Mamba, i.e., $p$ is FineWeb-Edu. All curves in Fig. 2 use FineWeb-Edu as the x-axis; the corresponding $E_{x|p}$ is given by the fit depicted in the left-most plot. $E_{y|p}$ for $y$ as The Pile UC and HellaSwag are given by the fits in the center and right plot, respectively.

*Table 2.* Architecture of the models we trained from scratch. All models generally use the tiktoken tokenizer. In addition to these models, for the tokenizer ablation §4.1, we train the largest Llama and Mamba models with the gpt2 tokenizer with beginning and end of sequence tokens.

| Architecture | Width | Depth | Parameters |
| --- | --- | --- | --- |
| Llama | 1024 | 12 | 416 M |
| Llama | 1024 | 8 | 365 M |
| Llama | 1024 | 4 | 314 M |
| Llama | 512 | 12 | 172 M |
| Llama | 512 | 8 | 158 M |
| Llama | 512 | 4 | 145 M |
| Llama | 256 | 12 | 76 M |
| Llama | 256 | 8 | 72 M |
| Llama | 256 | 4 | 59 M |
| Mamba | 1024 | 24 | 420 M |
| Mamba | 1024 | 16 | 367 M |
| Mamba | 1024 | 8 | 315 M |
| Mamba | 512 | 24 | 172 M |
| Mamba | 512 | 16 | 158 M |
| Mamba | 512 | 8 | 145 M |
| Mamba | 256 | 24 | 76 M |
| Mamba | 256 | 16 | 73 M |
| Mamba | 256 | 8 | 69 M |

*Table 3.* Details of the pretrained models used. All models do not use beginning and end of sequence tokens.

| Name | Pretraining data | Tokenizer Class | Parameters |
|---|---|---|---|
| GPT-J | The Pile | GPT2 | 6B |
| GPT-Neo | The Pile | GPT2 | {125M, 1.3B, 2.7B} |
| FineWeb Ablation Models | The Pile | GPT2 | 1.7B |
| FineWeb Ablation Models | C4 | GPT2 | 1.7B |
| FineWeb Ablation Models | FineWeb-Edu | GPT2 | 1.7B |
| GPT-NeoX | The Pile | GPT-NeoX | 20B |
| Pythia (small) | The Pile | GPT-NeoX | {70M, 160M, 410M} |
| Pythia (large) | The Pile | GPT-NeoX | {1B, 1.4B, 2.8B, 6.9B, 12B} |
| Mamba | The Pile | GPT-NeoX | {130M, 370M, 790M, 1.4B, 2.8B} |
| Mamba2 | The Pile | GPT-NeoX | {130M, 370M, 790M, 1.4B, 2.8B} |

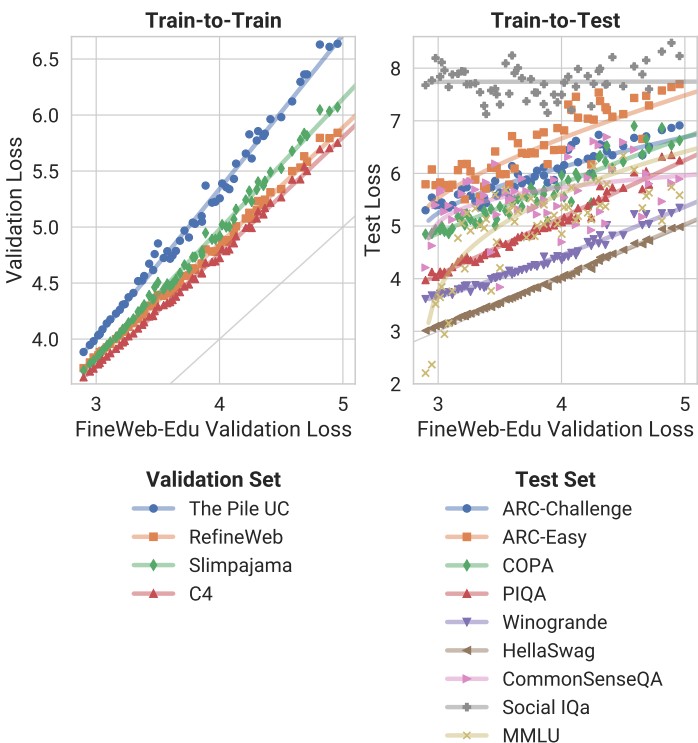

*Figure 11.* **Loss-to-Loss Scaling for FineWeb-Edu-trained Llama**.

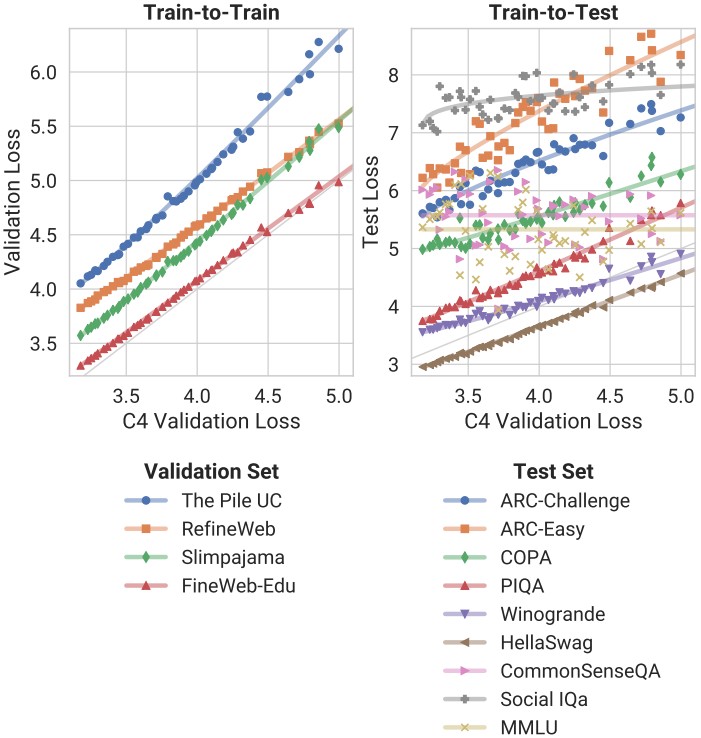

*Figure 12.* **Loss-to-Loss Scaling for C4-trained Llama**.

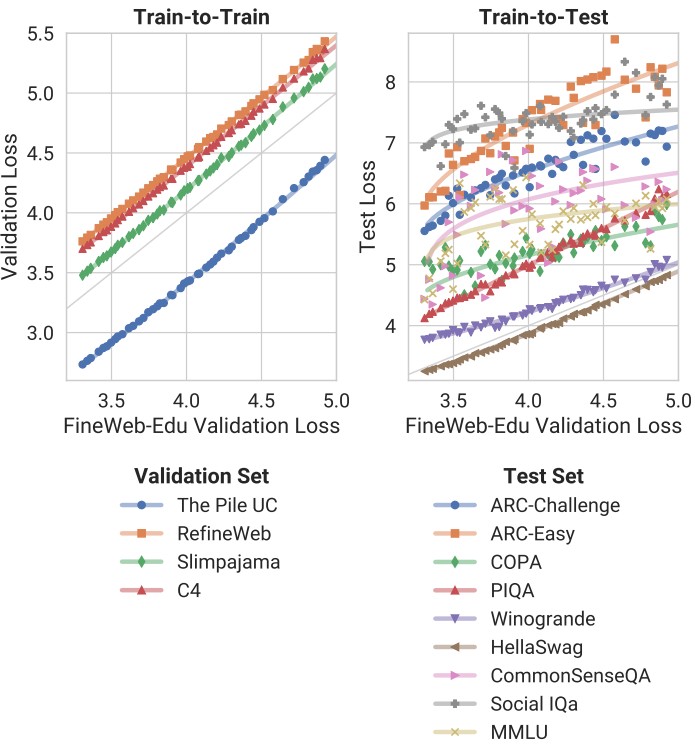

*Figure 13.* **Loss-to-Loss Scaling for The Pile-trained Llama**.

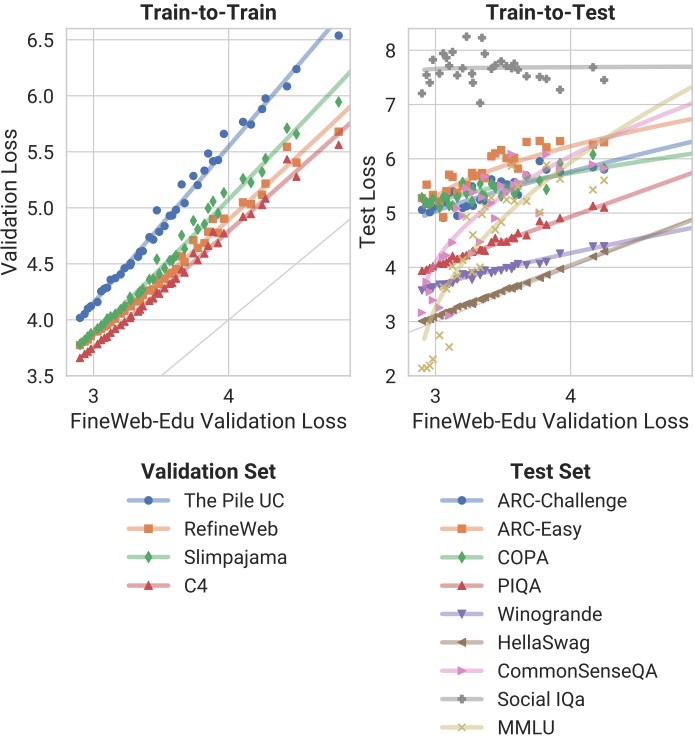

*Figure 14.* **Loss-to-Loss Scaling for FineWeb-Edu-trained Mamba**.

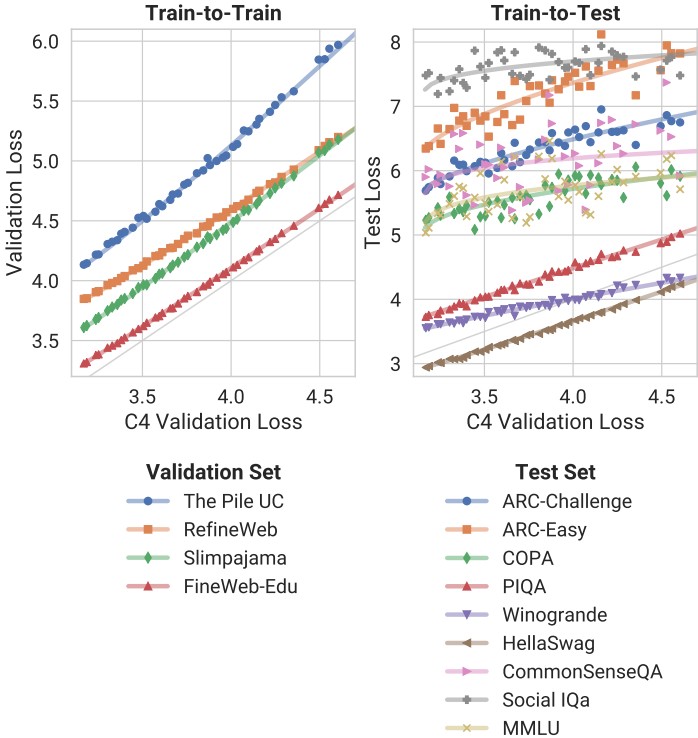

*Figure 15.* **Loss-to-Loss Scaling for C4-trained Mamba**.

*Table 4.* Area between fitted BPB curves for different interventions, evaluated on the interval [0, 2].

| | Base | Tokenizer | Data | Architecture | Optimizer | Size | Context |
|---|---|---|---|---|---|---|---|
| **Base** | — | 0.01 | 0.22 | 0.02 | 0.02 | 0.02 | 0.02 |
| **Tokenizer** | 0.01 | — | 0.19 | 0.03 | 0.03 | 0.03 | 0.03 |
| **Data** | 0.22 | 0.19 | — | 0.17 | 0.19 | 0.19 | 0.19 |
| **Architecture** | 0.02 | 0.03 | 0.17 | — | 0.02 | 0.02 | 0.02 |
| **Optimizer** | 0.02 | 0.03 | 0.19 | 0.02 | — | 0.00 | 0.01 |
| **Size** | 0.02 | 0.03 | 0.19 | 0.02 | 0.00 | — | 0.00 |
| **Context** | 0.02 | 0.03 | 0.19 | 0.02 | 0.01 | 0.00 | — |

## F. Intervention Results for Different Choices of x-Axis

We show variations of Figs. 4 to 6 from §4.1 with C4 validation loss as the x-axis in Figs. 17 to 19. Variations for Figs. 7 to 9 from §4.2 are shown in Figs. 20 to 22.

## G. Intervention Results without Averaging

The causal analysis in §4 was performed on scaling laws for *average* validation or test loss. Figs. 23 to 28 show illustrative results on scaling laws for individual datasets.

## H. Additional Train-to-Test Scaling Laws

We provide train-to-test scaling laws for the interventions performed in §4.2 and Figs. 7 to 9 in Figs. 29 to 34.

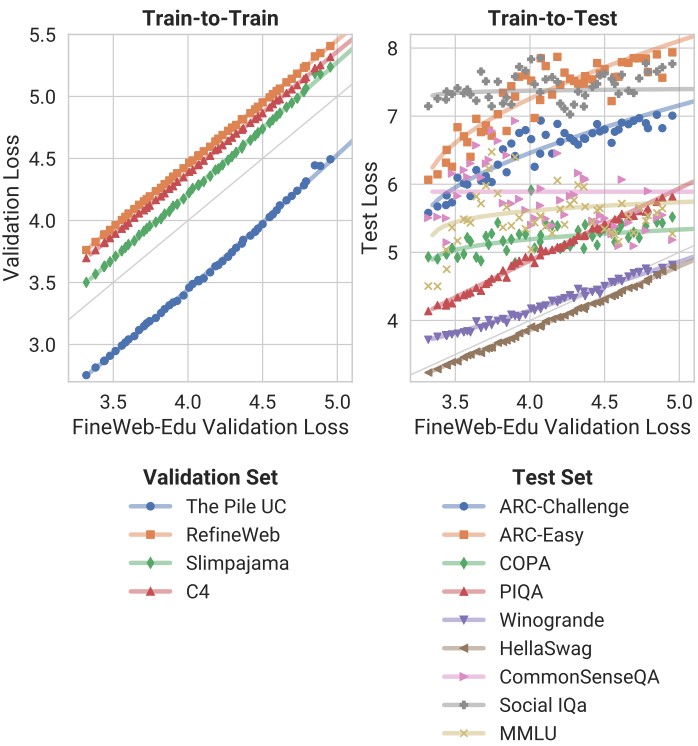

*Figure 16.* **Loss-to-Loss Scaling for The Pile-trained Mamba**.

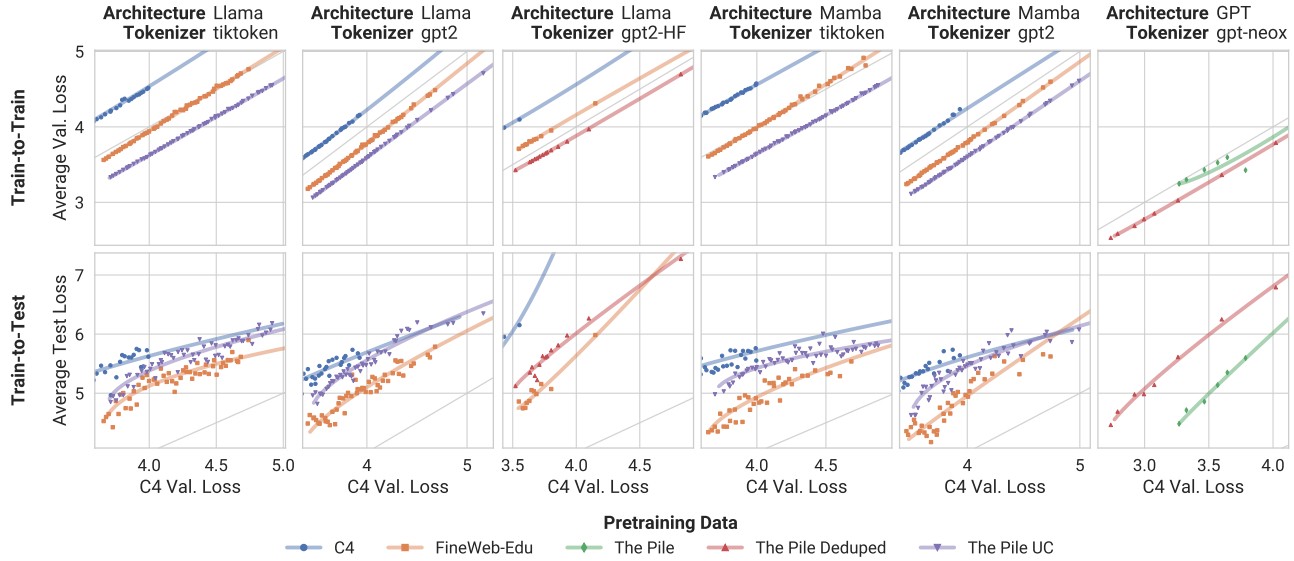

*Figure 17.* **Pretraining data has a substantial impact on loss-to-loss scaling laws**.

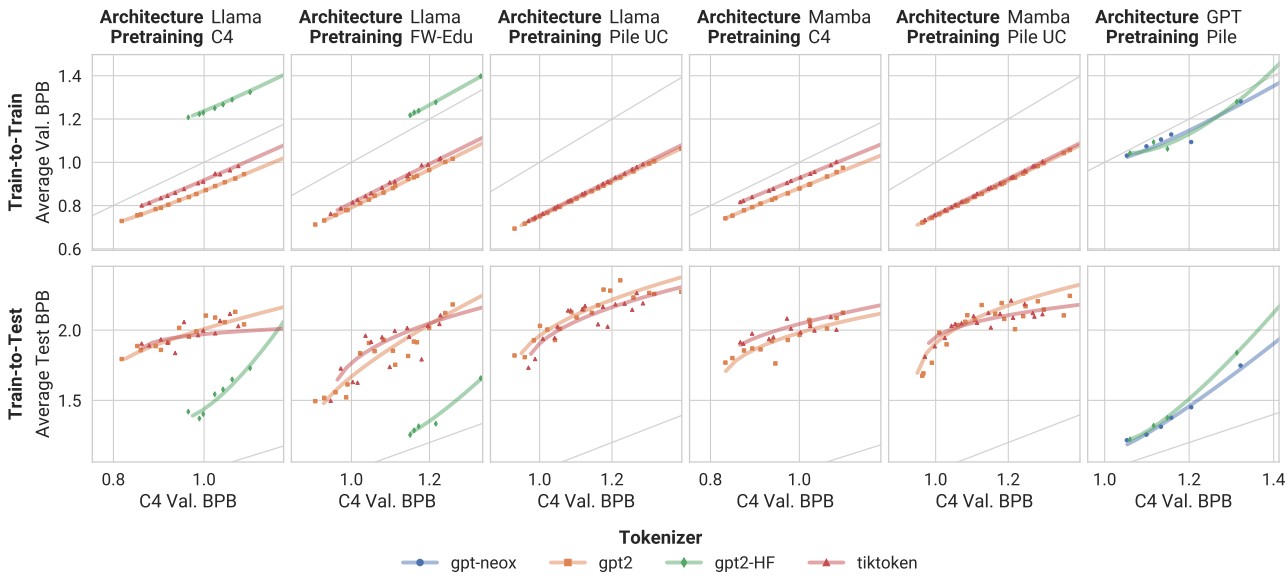

*Figure 18.* **The tokenizer has a minor impact on loss-to-loss scaling laws**.

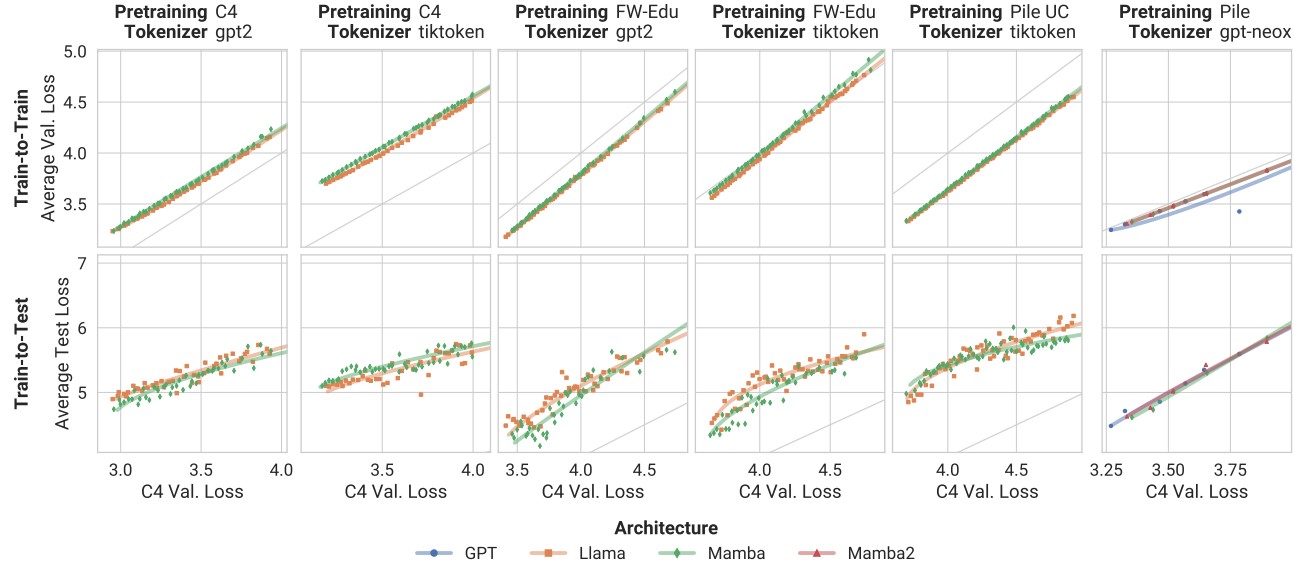

*Figure 19.* **Architecture has limited impact on loss-to-loss scaling laws**.

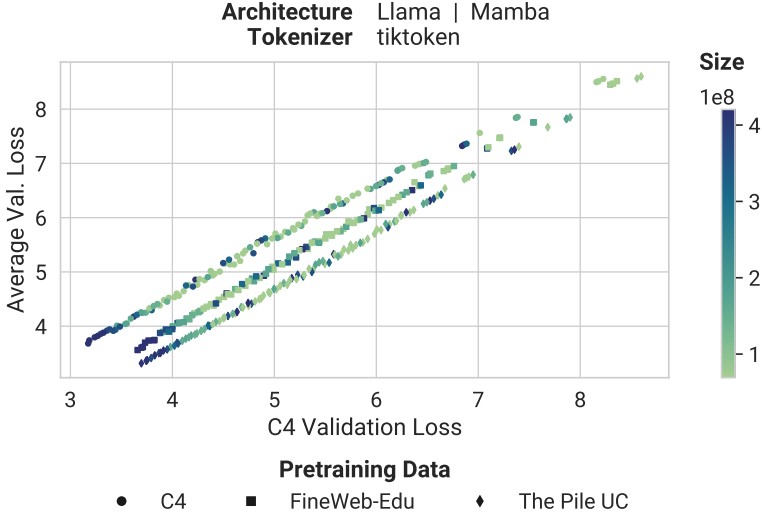

*Figure 20.* **Model size does not affect train-to-test scaling**.

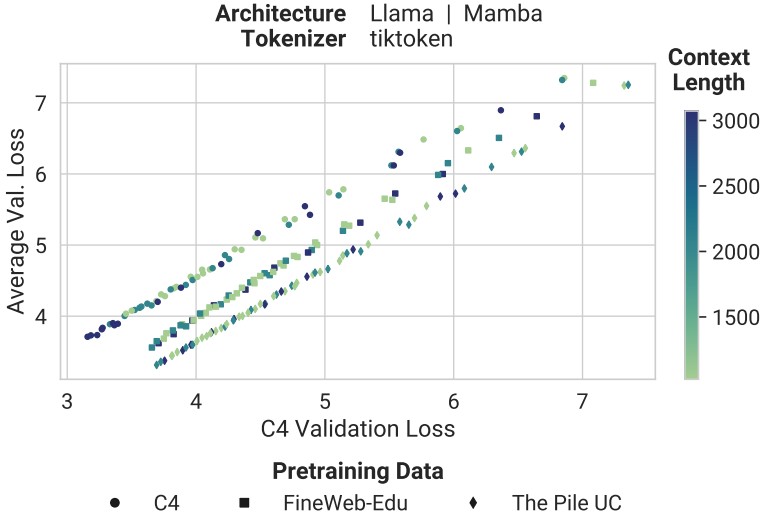

*Figure 21.* **Context length does not affect train-to-test scaling**.

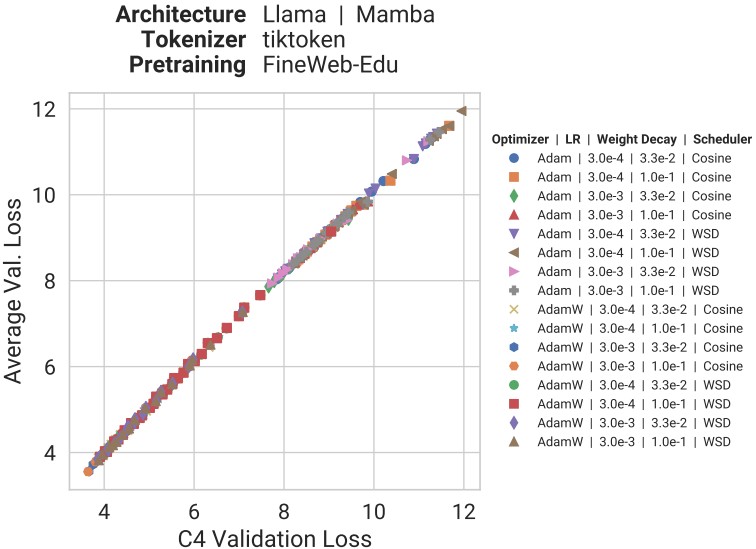

*Figure 22.* **Optimizer settings do not affect train-to-test scaling**.

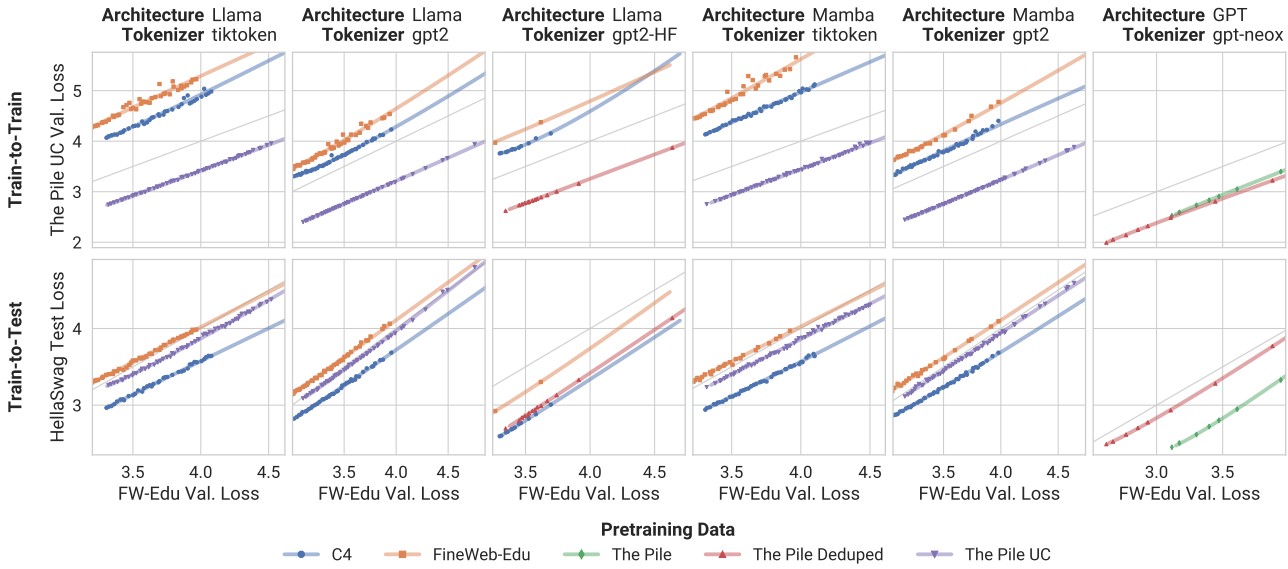

*Figure 23.* **Pretraining data has a substantial impact on loss-to-loss scaling laws**.

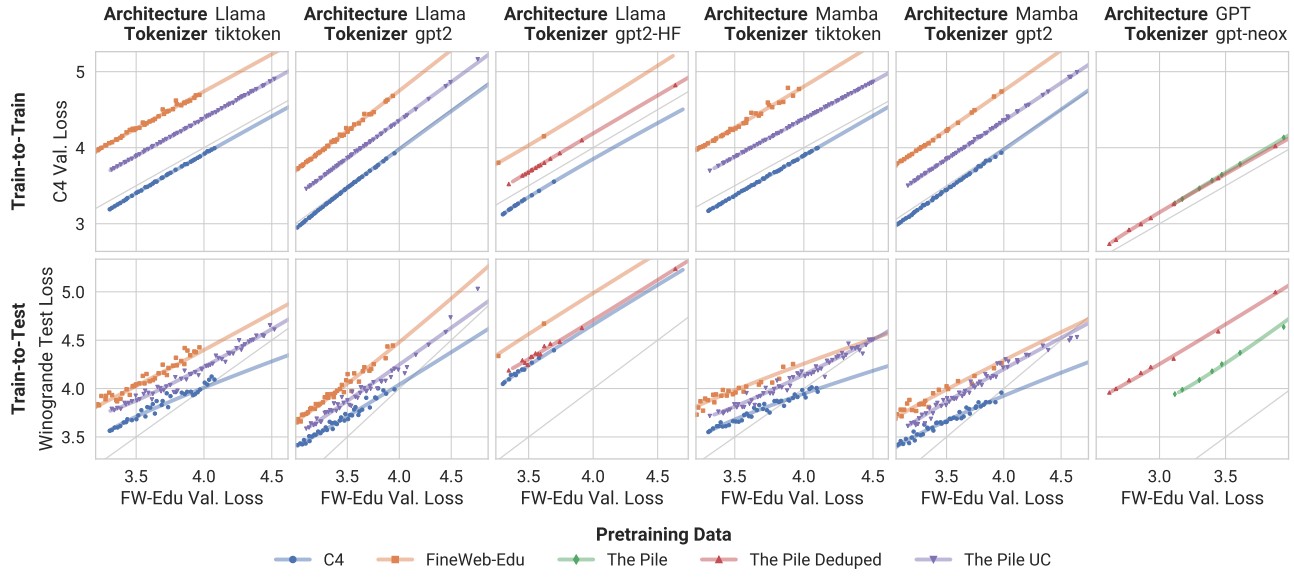

*Figure 24.* **Pretraining data has a substantial impact on loss-to-loss scaling laws**.

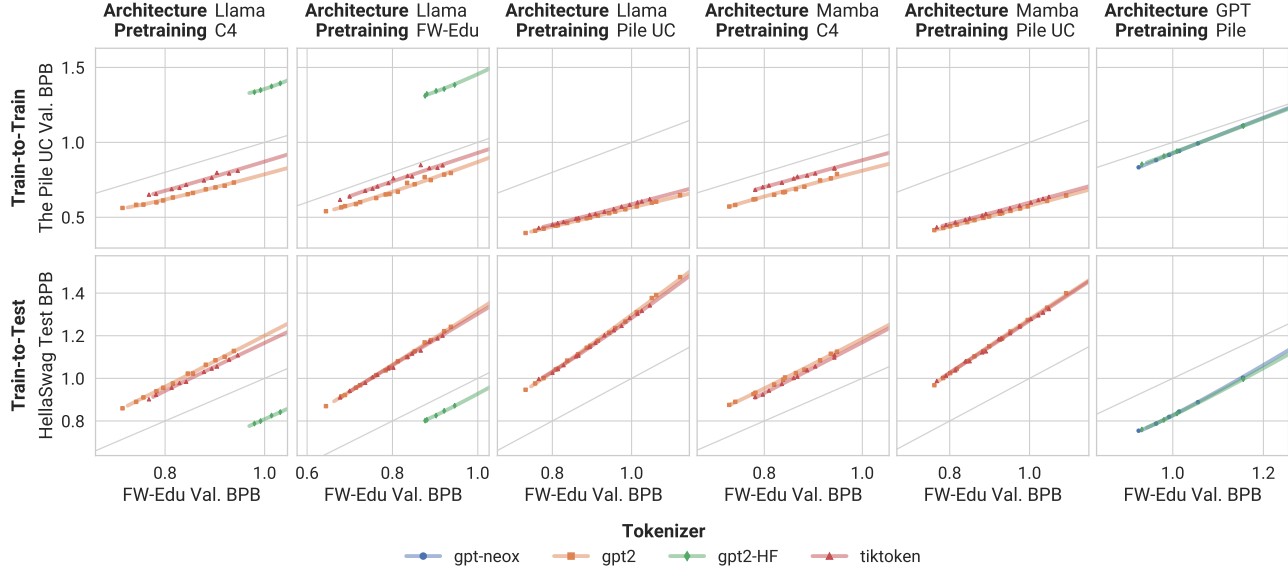

*Figure 25.* **The tokenizer has a minor impact on loss-to-loss scaling laws**.

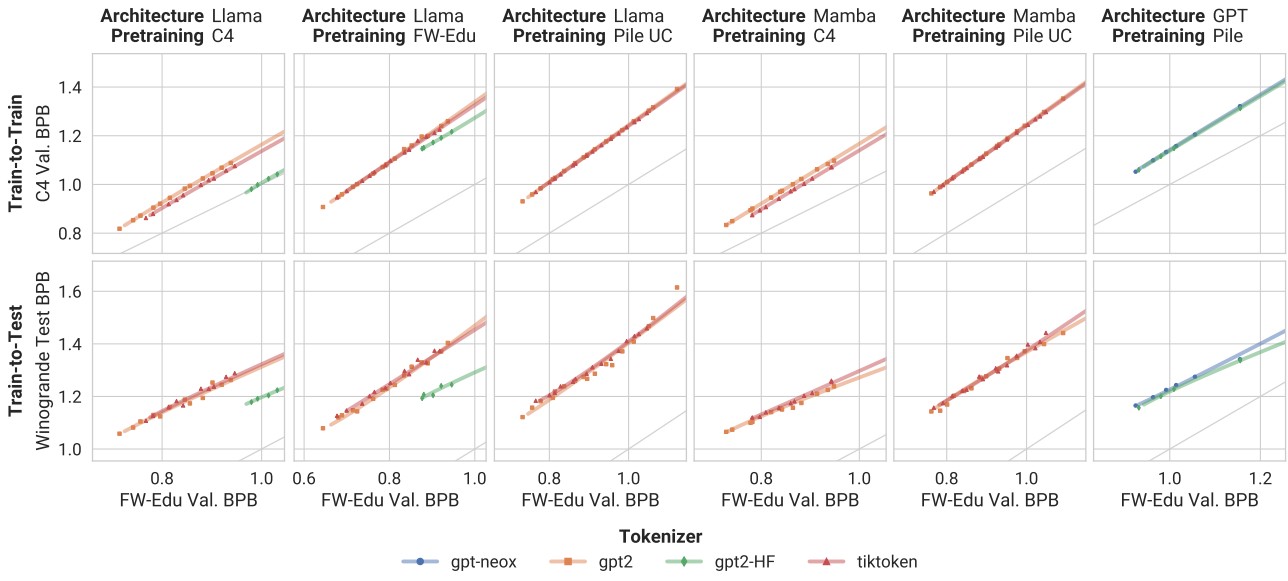

*Figure 26.* **The tokenizer has a minor impact on loss-to-loss scaling laws**.

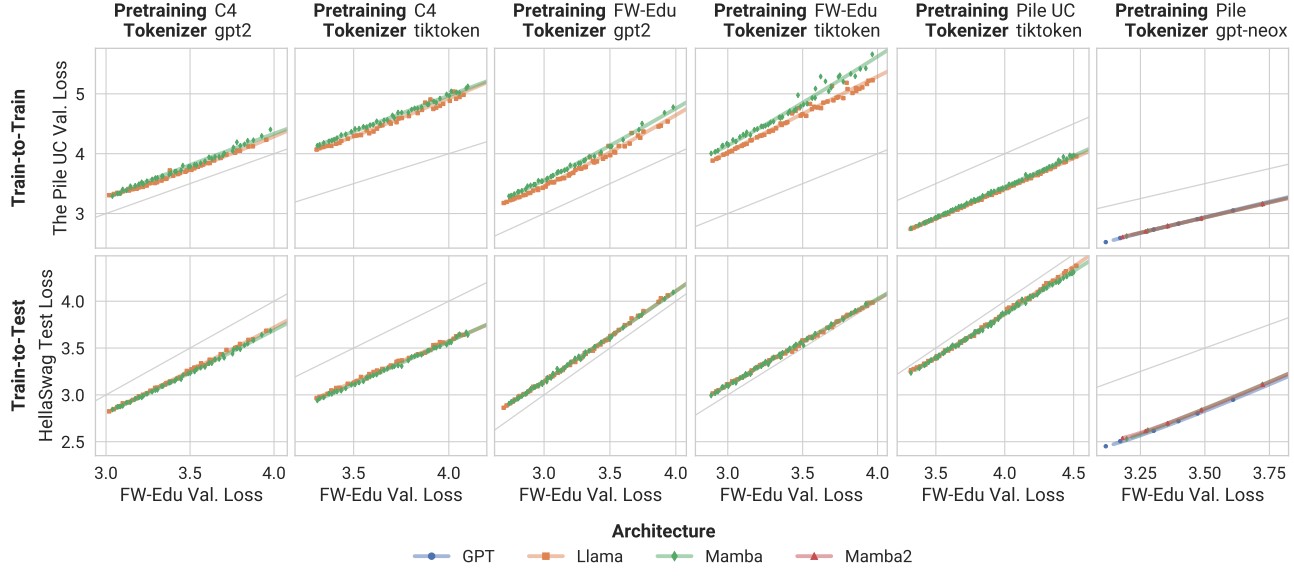

*Figure 27.* **Architecture has limited impact on loss-to-loss scaling laws**.

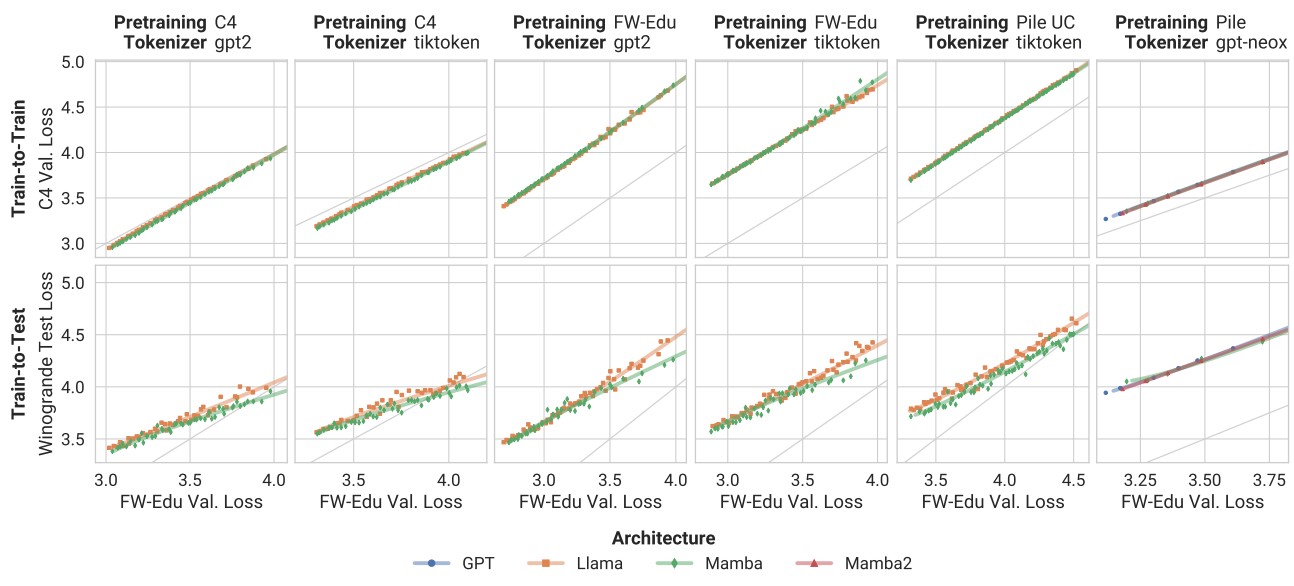

*Figure 28.* **Architecture has limited impact on loss-to-loss scaling laws**.

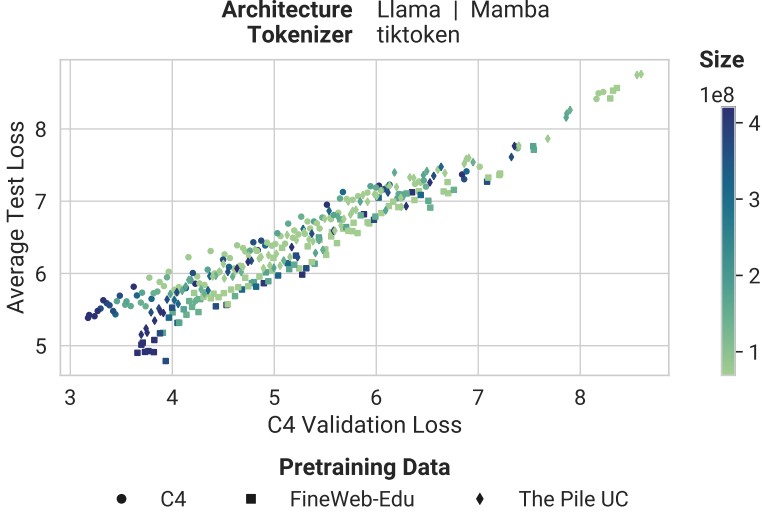

*Figure 29.* **Model size does not affect train-to-test scaling**.

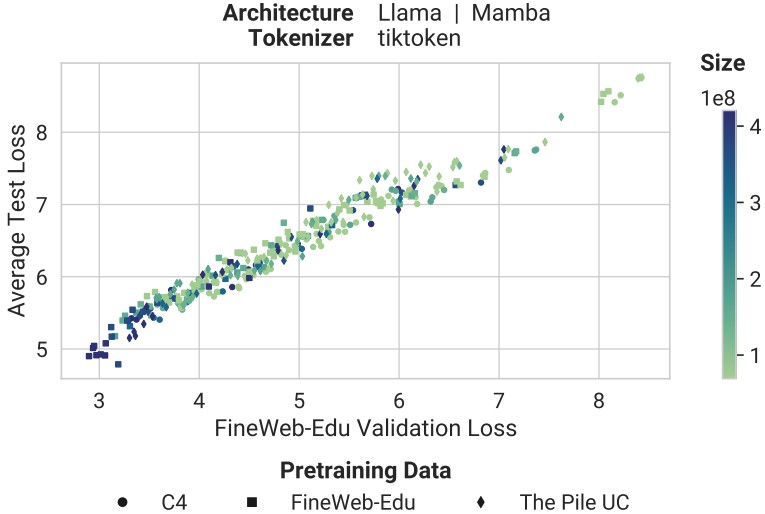

*Figure 30.* **Context length does not affect train-to-test scaling**.

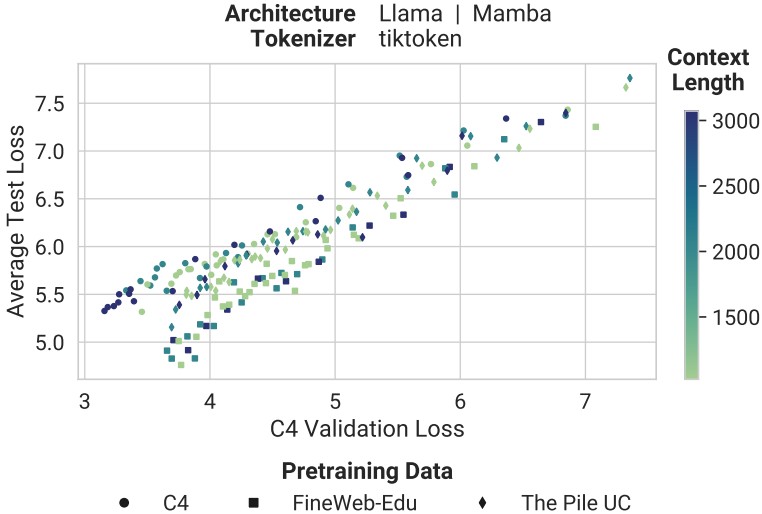

*Figure 31.* **Optimizer settings do not affect train-to-test scaling**.

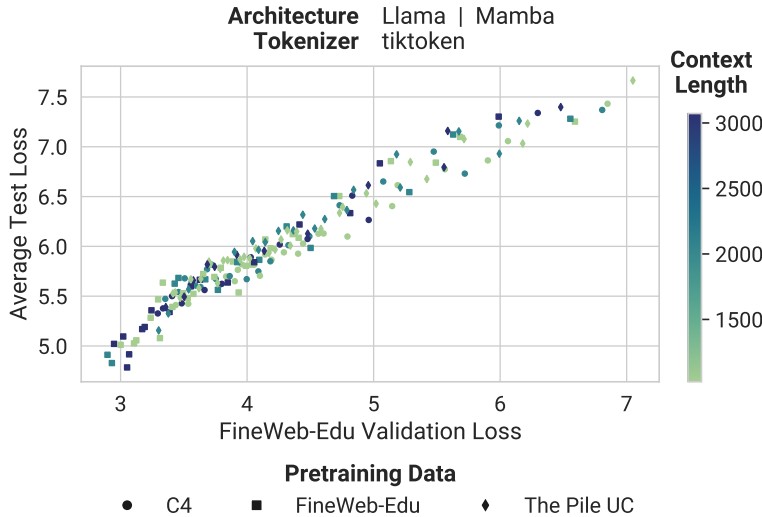

*Figure 32.* **Model size does not affect train-to-test scaling**.

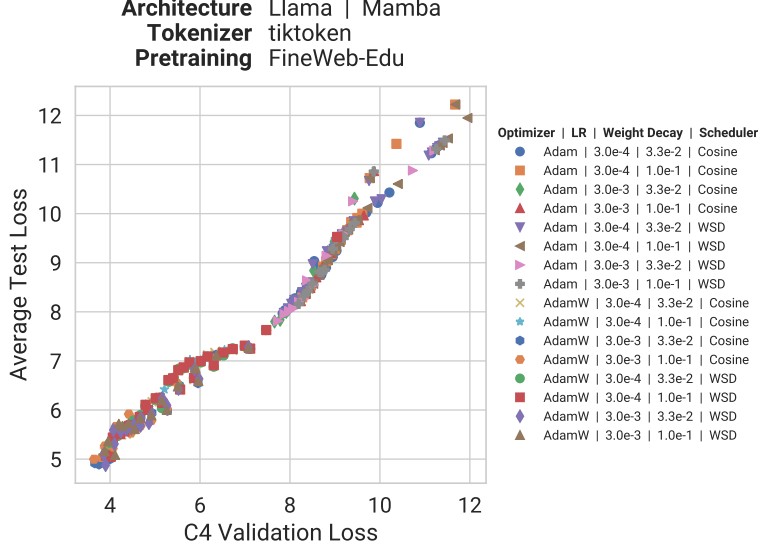

*Figure 33.* **Context length does not affect train-to-test scaling**.

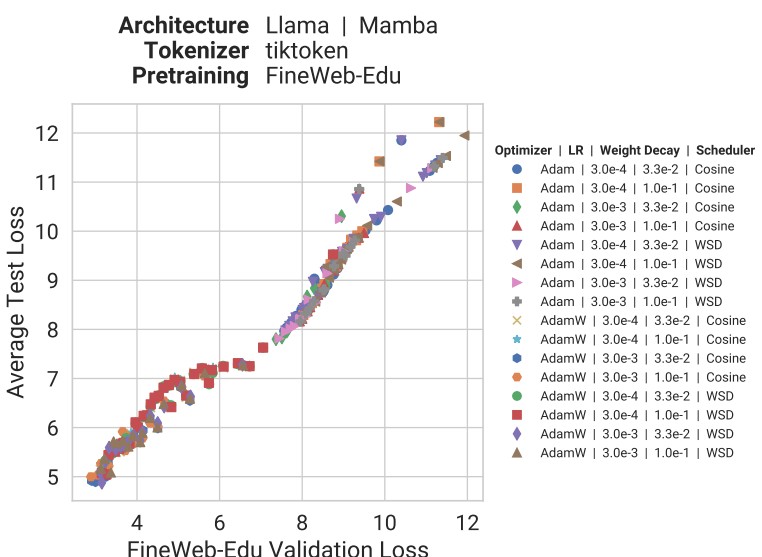

*Figure 34.* **Optimizer settings do not affect train-to-test scaling**.

