# OpenReview forum: "LLMs on the Line: Data Determines Loss-to-Loss Scaling Laws"
_ICML.cc/2025/Conference — ICML 2025 poster_

### Official Review · Reviewer_6jji · 2025-03-03

**Overall Recommendation:** 1

**Summary:**

The paper focuses on loss-to-loss scaling laws in large language models (LLMs), which relate losses across pretraining datasets and downstream tasks. The author finds that 1. loss-to-loss scaling consistently follows shifted power-law trends, enabling prediction of test performance from training loss, as detailed in the conclusion section. 2. Pretraining data and tokenizer are identified as the dominant factors shaping these scaling laws, with experiments showing significant impact when these are varied. 3. Architecture has limited influence and model size, context length, and optimization settings (e.g., Adam, cosine schedules) have negligible effects. 4. The authors recommend that practitioners should prioritize curating suitable pretraining datasets for optimal downstream performance, while architectures and other settings can be freely optimized for training efficiency.

## update after rebuttal
I am keeping my score since my concerns are not fully addressed.

**Claims And Evidence:**

Yes, each claim is accompanied by empirical results.

**Essential References Not Discussed:**

No.

**Experimental Designs Or Analyses:**

The experiment design is sound and the ablation study is thorough. However, the conclusions drawn are not quantitative, drawing on subjective judgments when deciding which factors affect scaling laws more.

**Methods And Evaluation Criteria:**

Yes, the authors train a series of models to validate their scaling laws.

**Other Comments Or Suggestions:**

The paper is well written.

**Other Strengths And Weaknesses:**

Strengths:
1. The authors conduct a comprehensive analysis of factors affecting loss-to-loss scaling.
2. The experiments are well organized and the results are clear.

Weakness:
1. The experiments do not bring to the table findings not implied by existing scaling law formulation. For instance, scaling law is a function of model size and data size. Changing the dataset would shift the scaling law by affecting the fitted parameters. Hence, there is a plethora of studies and open source efforts on producing higher quality datasets, whose scaling curve is different from the lower quality datasets (DCLM, redpajama, etc.)

2. Similarly, what the authors identified as less important is also inherently implied in existing scaling law formulation.

3. The author did not improve upon the existing scaling law's limitation, which is to connect training/validation loss to individual downstream task performance. For instance, if the authors believe that changing the architecture has a limited impact on loss-to-loss scaling laws, is it possible to show two models (one decoder-only transformer-based, one Mamba-based) with fixed recipes converge to similar downstream performance across tasks? This remains one of the most popular open questions in the field to this day.

**Questions For Authors:**

My questions are listed above.

**Relation To Broader Scientific Literature:**

The paper extends recent scaling law studies by systematically exploring factors influencing loss-to-loss scaling, contributing to understanding LLM performance optimization and generalization across tasks.

**Theoretical Claims:**

The paper is empirical.

---

> ### Author Rebuttal · Authors · 2025-04-01
>
> Thanks for your helpful comments and feedback.
>
> **[EDA1 - Conclusions are subjective]**: We now quantify our findings in two ways:
> 1. We quantify the goodness of fit of the loss-to-loss power laws as $R^2$. We show this in our revised [Fig. 1](https://ibb.co/kgpRZNZP) (note that Fig. 1 has also received other updates; please refer to our reply to Reviewer he3E). Generally, the goodness of fit is very close to $1$. We will similarly update all figures for the camera-ready.
> 2. We quantify the impact of different interventions as the area between fitted curves in a newly added [Tab. 2](https://ibb.co/7tjs1xL5). Pretraining data clearly has the biggest impact on the scaling laws. Please also refer to our reply to Reviewer he3E for our updated conclusion on the impact of the tokenizer.
>
> **[Conventional vs. loss-to-loss scaling laws]**: The distinction between compute-to-loss and loss-to-loss scaling laws may not have been sufficiently clear in the text and caused some confusion; we have updated our introduction and related work sections to explain this better.
> To reiterate, we focus on _loss-to-loss scaling laws_ (i.e., train-to-train, train-to-test, test-to-test), not _compute-to-loss scaling laws_ (e.g., Kaplan et al., 2020; Hoffmann et al., 2022). While compute-to-loss scaling laws are primarily used to find optimal compute budgets, loss-to-loss scaling laws can help study generalization, i.e., how performance transfers from training distributions to downstream tasks (Brandfonbrener et al., 2024; Du et al., 2025). Although compute-to-test scaling can be informative, they do not allow for connecting training loss to downstream performance, as you pointed out. This is where loss-to-loss scaling laws become a fascinating object of study, as they explicitly model how training/validation loss is converted to test loss, a proxy for test performance.
>  Please also see our response to reviewer fuqS on the utility of loss-to-loss scaling laws.
>
> **[W1, W2 - Findings are implied by existing scaling law formulations]**: We respectfully disagree. As the reviewer points out correctly, conventional scaling laws (i.e., compute-to-loss scaling laws) are a function of model and data size. That formulation alone does not automatically imply that changing data _quality_ without affecting data _size_ should impact the scaling law.
>
> Having said that, for _compute-to-loss scaling laws_, it is of course well known that data quality _does_ play a role. However, other factors like architecture, optimizer, and learning rate scheduler also matter for compute-to-loss scaling laws (Li et al., 2025). It is not at all “inherently implied in existing scaling law formulations” which factors impact scaling laws and which don’t – after all, none of these factors are quantifiable and explicitly enter the scaling law.
>
> Now, _loss-to-loss scaling laws_ are a relatively recent object of study. Prior to our study (Takeaway 1), it was unclear whether loss-to-loss scaling laws could generally be described by shifted power laws at all, since previous studies like Brandfonbrener et al. are very limited in their settings. On top of that, it is unclear which factors, if any, play a role. Again, it is not at all “inherently implied in existing scaling law formulations” whether pretraining data (quality, not size), architecture, tokenizer, or other hyper parameters might impact the scaling law. If we were to go by the experience of compute-to-loss scaling laws, we might assume that most of these factors matter. In contrast, we find that loss-to-loss scaling laws are notably insensitive to many factors, while strongly depending on the pretraining data. As reviewer EE76 points out, “this is of broader interest to the community, in terms of advancing our understanding of the role of architecture, optimization, etc. in downstream properties”.
>
> **[W3b - Mamba & Llama perform similarly]**: As you note, determining whether drastically different architectures like transformer-based Llama and state-space-based Mamba achieve similar downstream performance given identical training setups is a significant open question.
>
> We have to emphasize the distinction between compute-to-loss and loss-to-loss training laws. While compute-to-loss training laws can answer whether two models converge to the same training loss with increasing compute, loss-to-loss training laws answer whether models reach the same downstream performance given a training loss.
>
> In this sense, yes, it is precisely the case that Mamba and Llama models “converge to similar downstream performance across tasks” given they reach the same training loss. To further illustrate this, we present in [Tab. 3](https://ibb.co/tpgMmP1Z) multiple Llama and Mamba models trained under identical conditions, which show nearly indistinguishable downstream performances.
>
> We hope this addresses your concerns and encourages you to raise your score.
>
> (Li et al, 2025) (Mis)Fitting: A Survey of Scaling Laws

---

### Official Review · Reviewer_he3E · 2025-03-15

**Overall Recommendation:** 4

**Summary:**

The paper studies loss-to-loss scaling laws in language models, covering both predicting the language modeling log loss across different data distributions (“train-to-train”) and predicting log loss proxies for downstream tasks performance (“train-to-test’). The main finding in this paper is that loss-to-loss scaling is insensitive to model architecture and optimization hyperparameters, mildly sensitive to the choice of tokenizer, and very sensitive to the training data distribution.

## Update after the rebuttal

The authors have adequately addressed my concerns. I therefore increase my score to 4.

**Claims And Evidence:**

The claims of this paper are generally well supported by evidence, except for one notable issue: it is not clear if the language models studied in this paper can be called “large.” The authors train models of up to 420M parameters and supplement the analysis with additional models of undisclosed sizes. Based on the FW-Edu validation loss, none of the supplementary models are large: using their own models the authors reach a loss of about 2.9 (according to Fig. 30) while the lowest loss shown in the paper is about 2.7 (in Fig. 1), which makes me unsure if even the 1.8B parameter FineWeb-Edu ablation model by HuggingFace was included.

Clearly, truly large models (say with 7B parameters or more) were not included in this study, and I am not sure why: the web is full of scaled families of open-weights LM’s, some including intermediate checkpoints and trained on fixed datasets. These models could be downloaded, evaluated, and used to test the claims of the paper on actually large language models.

The lack of evaluation on larger models is my primary concern regarding this paper - if it is addressed I will be happy to increase my score.

**Essential References Not Discussed:**

None to my knowledge.

**Experimental Designs Or Analyses:**

See “Claims And Evidence” above.

**Methods And Evaluation Criteria:**

See “Claims And Evidence” above.

**Other Comments Or Suggestions:**

1. I had a hard time parsing what is meant by “train-train” and “train-to-test” - I suggest including a more detailed and self-contained explanation in the beginning of the revised paper.

2. When comparing different tokenizers, comparing the average negative log probability per token is not entirely appropriate, since different tokenizers have different compression efficiencies and hence represent the same data using a different number of tokens. Instead, you should normalize the negative log-probability by a tokenizer-independent quantity, such as the number of bytes of uncompressed text in the sequence, or the number of tokens used by a fixed tokenizer. It would be interesting to see if such improved reckoning moves the “lines” corresponding to different tokenizers closer.

**Other Strengths And Weaknesses:**

N/A.

**Questions For Authors:**

Precisely which models were evaluated and what were the evaluation results? You should answer this by tabulating the results of your entire testbed and include it as supplementary material. See “Claims And Evidence” above for context.

**Relation To Broader Scientific Literature:**

This paper contributes to the body of evidence that the performance of neural networks under distribution shift tends to be predictable with very few “effective robustness” interventions beyond changing the training data

**Theoretical Claims:**

N/A.

---

> ### Author Rebuttal · Authors · 2025-04-01
>
> We thank you for your helpful comments and feedback. We address each of your questions and concerns below.
>
> **[Q1 - Models trained and evaluated and complete results]**: Thank you for noticing this; we have made amendments in multiple places: First, we now mention in Sec. 4 the size of not only our models, but also the models sourced from HuggingFace. Second, we have expanded on App. A and Tab. 1 to clarify the models we trained and created a new table for models we evaluated from HuggingFace (see [Tab. 2](https://ibb.co/xq3KgQ4N)). Third, we will release the entire data frame with all model settings and evaluation results for the roughly 6000 checkpoints and our complete code with the camera-ready version.
>
> Here’s an abridged overview of the evaluated models:
>
> - We trained Llama and Mamba models ranging from approximately 60M to 420M parameters from scratch. We have also added some 7B models now, see below.
>  - We evaluated pretrained models from HuggingFace: Pythia (70M, 160M, 410M, 1B, 1.4B, 2.8B, 6.9B, 12B), GPT-Neo models (125M, 1.3B, 2.7B), GPT-NeoX (20B), and GPT-J (6B), all trained on variants of The Pile. These models are included in the right-most columns of Figs. 4–6, 16–18, and 22–27.
> - We also evaluated FineWeb ablation models (1.7B) from HuggingFace trained on The Pile, C4, and FineWeb-Edu. These models are also included in Figs. 4-6 (e.g. Fig. 4 column 3).
>
> Note that although we fit scaling laws using all available checkpoints, we display a randomly selected subset in figures for readability. We have made this more evident in the text and figure captions.
>
> **[CE1 - Training/Evaluating large models]**: As explained above, our analysis already includes publicly available pretrained models with up to 1.7B parameters for Llama, up to 20B parameters for GPT, and up to 2.7B parameters for Mamba. We apologize for this not being evident in the original manuscript. As illustrated in Figs. 4-6 (especially in the right-most column showing the largest GPT models), these models generally achieve lower loss on validation and test sets compared to our models trained from scratch, but still adhere to the power-law trends and confirm our key insights. For example, the right-most column in Fig. 6 contains all of the largest HuggingFace models and clearly demonstrates that architecture does not affect loss-to-loss scaling laws even at these scales.
>
> Following your suggestion, we revisited available pretrained models. However, most other publicly available checkpoints of large scale models (≥7B) are unsuitable for our analysis as they are trained on (often undisclosed) data mixtures that prevent us from directly comparing them to other models (.e.g, the Falcon series).
>
> All that said, we agree that especially the impact of the pretraining data in Fig. 4 is not conclusively shown for larger models, since the largest HuggingFace models included in our analysis were all trained on variants of The Pile (see Fig. 4, right-most column). To remedy this, we have now trained additional 7B models for 1B tokens ourselves. Due to compute and time constraints we have limited this to three settings:
>
>  - Llama-7B (tiktoken tokenizer) on FineWeb-Edu / on C4
>  - Llama-7B (GPT-2 tokenizer) on FineWeb-Edu
>
> We’ve included all these models in our revised [Fig. 1](https://ibb.co/kgpRZNZP). All models follow the established power-law scaling curves and confirm our main conclusions.
>
> **[OCS1 - Clarifying taxonomy used in paper]**: Thanks for the suggestion. We will state and clarify the taxonomy used in the paper early in the manuscript. We have also added an appendix section that explains the scaling laws in more detail, which we describe in our reply to reviewer EE76.
>
> **[OCS2 - Normalizing tokenizer comparisons]**: Thank you for this excellent suggestion! We performed the experiment and have updated [Fig. 1](https://ibb.co/kgpRZNZP) for BPB. Indeed, after normalizing negative log-probability by bytes, the previously distinct tokenizer lines collapse onto each other. This confirms even more strongly that architecture, tokenizer, and optimization hyperparameters minimally affect loss-to-loss scaling—only pretraining data distribution truly matters. We will update other figures and sections of the manuscript accordingly.
>
> We hope this addresses your concerns fully and encourages you to raise your score.

---

> > ### Comment · Reviewer_he3E · 2025-04-05
> >
> > Thank you for your response, and it is encouraging to see that normalizing by the tokenizer efficiency makes the choice of tokenizer even less significant.
> >
> > However, my main concern is not only about the number of parameters in the model considered; it is about seeing how well the proposed scaling laws persist as we push the loss further down.
> > * Given a model family, I am not sure I follow why it matters if the training mixture is disclosed or not: as long as it is consistent across the models in the family, the test losses should be “on a line.”
> > * Moreover, some works provide model families with completely disclosed training data, and covering a larger range of compute than currently considered. One example that I’m aware of is the models released as part of “Language models scale reliably with over-training and on downstream tasks” by Gadre et al. (2024).
> > * I am particularly curious about the 1.7B ablation model trained on FineWeb-Edu. Is this model included in Figure 1? Could you write here its FineWeb-Edu validation loss and its HellaSwag test loss?

---

> > > ### Author Response · Authors · 2025-04-08
> > >
> > > Thank you for explaining your concerns in more detail. It seems we have misunderstood why you were asking about larger models. Specifically, there are two questions to consider:
> > >
> > > 1. Do interventions have the same effect (pretraining data matters, other factors mostly don’t) for larger model sizes?
> > > 2. Do larger models with lower loss still follow the power-law formulation we use?
> > >
> > > We believe that our previous reply has shown conclusively that point 1 holds, but understand now that you’re asking about point 2, which we address below:
> > >
> > >  - **[Using different model families]**. Our argument was specific to point 1. If we want to perform targeted interventions, we need to of course be able to match the model in all but one of pretraining data, architecture, tokenizer, etc. This is why our original comparison figures only include model families that we can compare to at least one other model family. That said, you are right of course that for point 2, any model family with sufficient checkpoints can be used. We now additionally evaluate an OLMo model with up to 7B parameters that achieves low loss.
> > >  - **[Scaling law fits for larger models]**. To show that point 2 holds, we show a [new figure](https://ibb.co/93N3vtD2) containing scaling law fits for models up to 21B parameters and with overall lower loss. We will add this figure in the appendix.
> > > The figure contains our own Llama models up to 7B parameters (green diamonds), GPT models (Pythia/GPT-NeoX) from HuggingFace up to 21B parameters (blue circles and orange squares), the HuggingFace FineWeb-Edu Ablation Model (red triangles), and the newly evaluated OLMo model (purple triangles). Note that the HuggingFace models have always been part of our analysis, as stated in our previous reply.
> > > Kindly keep in mind that the models in this figure cannot directly be compared as in our interventional study, since they differe along multiple axes, sometimes in subtle ways. E.g., the HF ablation model (red triangles) uses a different and undisclosed version of FineWeb-Edu compared to our Lllama model (green diamonds).
> > > Multiple of these models achieve significantly lower test loss than our own Llama models, and all models follow a power law.
> > >  - **[Low-loss regime]**. With the HuggingFace GPT models and the newly-evaluated OLMo, our analysis includes models in the sub-2.5 Loss regime on Hellaswag (sub-0.8 BPB).
> > > We also note that we fit scaling laws following the methodology of Brandfonbrener et al. (2024). While their setting is limited in terms of architectures, tokenizers, etc., they show that loss-to-loss scaling laws are predictive to large model size / low loss. We believe that the whole of their and our results leaves little doubt that loss-to-loss scaling laws persist into the low-loss regime.
> > >  - **[Data for the 1.7B FineWeb-Edu HuggingFace ablation model]**. This model is not included in Fig. 1 as it differs from our Llama models in multiple dimensions (tokenizer, pretraining data, as explained above). It is, however, included in Fig. 4, column 3 and Fig. 5, column 2 and their variants in the appendix as “Llama, gpt2-HF, FineWeb-Edu”. Its validation and test loss is included in the complete dataframe of all checkpoints that we will release with our paper. For your convenience, we show an excerpt below.
> > >
> > > | Steps | HellaSwag Loss | FW-Edu Loss | HellaSwag BPB | FW-Edu BPB |
> > > |------:|---------------:|------------:|--------------:|-----------:|
> > > |   2k | 3.30 | 3.62 | 0.96 | 0.85 |
> > > |  10k | 2.92 | 3.26 | 0.85 | 0.77 |
> > > |  20k | 2.84 | 3.18 | 0.82 | 0.75 |
> > > |  30k | 2.80 | 3.14 | 0.81 | 0.74 |
> > > |  40k | 2.77 | 3.12 | 0.80 | 0.73 |
> > > |  50k | 2.76 | 3.10 | 0.80 | 0.73 |
> > > |  60k | 2.74 | 3.09 | 0.79 | 0.73 |
> > > |  70k | 2.73 | 3.07 | 0.79 | 0.72 |
> > > |  80k | 2.72 | 3.06 | 0.79 | 0.72 |
> > > |  90k | 2.70 | 3.05 | 0.78 | 0.72 |
> > > | 100k | 2.69 | 3.03 | 0.78 | 0.71 |
> > > | 110k | 2.69 | 3.02 | 0.78 | 0.71 |
> > > | 120k | 2.67 | 3.01 | 0.77 | 0.71 |
> > > | 130k | 2.66 | 3.00 | 0.77 | 0.71 |
> > > | 140k | 2.66 | 2.99 | 0.77 | 0.70 |
> > > | 150k | 2.65 | 2.98 | 0.77 | 0.70 |
> > > | 160k | 2.65 | 2.98 | 0.77 | 0.70 |
> > >
> > > We hope that this resolves your remaining concerns. We thank you for your valuable suggestions that led to multiple additions and improvements of the manuscript, and kindly ask that you consider adjusting your score in light of this.

---

### Official Review · Reviewer_EE76 · 2025-03-20

**Overall Recommendation:** 3

**Summary:**

This paper explores how loss-to-loss scaling laws depend on various factors in the training setting. While compute-to-loss scaling laws are often studied (i.e., training on X tokens, Y parameters will give you Z loss), there is recent interest in loss-to-loss scaling laws, which show how evaluation/training on one dataset can translate into evaluation/training on another. This paper finds that by varying the pre-training dataset, the relationship between validation loss and downstream test loss can change significantly. Varying the tokenizer can also change the scaling trend, while things like the model architecture (like Llama vs Mamba), optimization hyperparameters, context length, and model size have little impact.

**Claims And Evidence:**

Evidence from numerous empirical runs supports the claim that data primarily determines loss-to-loss scaling, while other factors (optimization, architecture, etc.) impact this scaling less.

**Essential References Not Discussed:**

None.

**Experimental Designs Or Analyses:**

Experimental design and analyses appear sound, although there could be more clarity on how the shifted power law is fit (i.e., how to compute $E_{x|p}, E_{y|p}$ and parameters $K$ and $\kappa$).

**Methods And Evaluation Criteria:**

The evaluation criteria makes sense for the problem at hand.

**Other Comments Or Suggestions:**

None.

**Other Strengths And Weaknesses:**

Weaknesses:


Clarity:
- It is slightly unclear how a train-to-train scaling law is defined. In Brandfonbrener (2024), there are *two* models being trained on different data with the same number of params and tokens, and that makes up the x-axis vs y-axis. However, in Figure 4 each point is derived from one model trained on a particular dataset, and evaluated on FineWeb-Edu as well as other validation domains (and averaged). I see that lines 156-160 briefly mention that each point in the paper's plots shows losses of one model; however, train-to-train and train-to-test should still be formally defined.
     - The clarity of the paper could be improved by explaining what each point on the various loss-to-loss plots represents, and concretely describing the experimental setup for each type of scaling law in the Appendix.
     - The clarity could also be improved by adding more description to the x and y axes in the plots. For instance, in Brandfonbrener et. al., they often write "Loss on Hellaswag (Trained on data 1)" and "data 1" is depicted by the colors and the legend. This would make it easier for the reader to understand what the different colored points mean in relation to the axes.

Significance:
- The paper needs to go one level deeper in discussing the practical applications of their findings. For instance, how can a practitioner, who is using one training dataset/one validation dataset, use the insights from this paper to better predict performance on another dataset/task? Moreover, in section 5, the paper says that the data distribution is the key for achieving a desirable loss-to-loss scaling and in turn achieve a great downstream performance---what defines "desirable" here?

**Questions For Authors:**

1. Can you clarify how the shifted power laws are fit? i.e., how to compute $E_{x|p}, E_{y|p}$ and parameters $K$ and $\kappa$.
2. How can we better understand the role of pre-training data in the loss-to-loss scaling law?
3. Improve clarity: can you precisely explain how train-to-train and train-to-test scaling laws were constructed?
4. Add more discussion of the practical applications of their findings - how can a practitioner use the insights from this paper? What defines a desirable loss-to-loss scaling that can be used to achieve better downstream performance?

**Relation To Broader Scientific Literature:**

Strengths:
+ This paper discovers that loss-to-loss scaling is not significantly impacted by changes in model architecture, model size, context length, and optimization settings. It is especially interesting that the scaling is not impacted by model architecture (Llama vs Mamba), which suggests that different architectures can converge to similar representations. This is of broader interest to the community, in terms of advancing our understanding of the role of architecture, optimization, etc. in downstream properties.
+ The implications of pre-training data determining the loss-to-loss scaling laws are interesting, because many data curation strategies primarily focus on optimizing one validation dataset's loss (for example, [1]). This paper suggests that we should exercise caution in this practice---achieving X loss on just one validation dataset could mean various different performances on a downstream task. That is, one validation dataset is not comprehensive enough to determine downstream performance.

Weaknesses:
- While this paper has the above interesting implication, it feels like it falls short in understanding how pre-training data alters the loss-to-loss scaling law. That is, if there is this extra confounding factor---the pretraining dataset---in these loss-to-loss scaling laws, how do we model this factor and eliminate the confounding? My hunch is that a multi-loss to loss scaling law could result in things lying on the same line (for example, see Data Mixing Laws [2]). The paper would have broader impact if it could provide a path forward for consistent loss-to-loss scaling laws, or how to make sense of pre-training data impacting these laws.

[1] https://arxiv.org/abs/2407.01492
[2] https://arxiv.org/abs/2403.16952

**Theoretical Claims:**

Not applicable.

---

> ### Author Rebuttal · Authors · 2025-04-01
>
> Thank you for your helpful comments and feedback. We address your concerns below:
>
> **[Q1, Q3, OSW Clarity 1 - On the construction and fitting of scaling laws]**: We have added two Appendix sections that (1) detail the scaling law formulation and (2) explain how parameters are estimated. Here’s a slightly abridged version:
>
> > **Scaling Law Details**
> We adopt the compute-to-loss scaling law formulation from (Brandfonbrener et al.) Eq. (4):
> > $$L\left(f^{(N, D)}\right) = E + \left( \left( \frac{A}{N} \right)^{\frac{\alpha}{\beta}} + \frac{B}{D} \right)^\beta,$$
> > where $f^{(N, D)}$ is a model with $N$ parameters trained on $D$ tokens and $E, A, B, \alpha, \beta$ are parameters to be fit.
> > Notably, the irreducible error $E$ captures the minimum loss possible for model $f$ in the limit of infinite model and data size.
> >
> > By default, this is fit using the training or validation loss.
> > However, as demonstrated by (Brandfonbrener et al.) and our experiments, we can alternatively predict the loss $L_x$ on dataset $\mathcal D_x$ achieved by model $f_p^{(N, D)}$ trained on the pretraining set $\mathcal D_p$:
> > $$L_x\left(f_p^{(N, D)}\right) = E_{x|p} + \left( \left( \frac{A}{N} \right)^{\frac{\alpha}{\beta}} + \frac{B}{D} \right)^\beta.$$
> > As in (Brandfonbrener et al.) Eq. (7), the irreducible error $E_{x|p} = L_x(f_p^*)$ then captures the minimum possible loss on $\mathcal D_x$ of a model trained on $\mathcal D_p$.
> >
> > With that, we can formulate the loss-to-loss scaling law for arbitrary combinations of pretraining data and two test or validation sets, as stated in Eq. (1).
>
> > **Fitting Details**
> > For each line in a plot corresponding to a loss-to-loss scaling law from Eq. (1), we first fit the two compute-to-loss scaling laws $L_x(f_p^{(N, D)})$ and $L_y(f_p^{(N, D)})$.
> > This yields estimates for the irreducible errors $E_{x|p}, E_{y|p}$, which correspond to the minimum x- and y-value of the loss-to-loss line.
> > We use SciPy's default `curve_fit` optimizer for fitting.
> > In rare cases when all checkpoints have the same number of parameters $N$ or same number of tokens $D$ (this is the case only for a small subset of the HuggingFace models) and a compute-to-loss scaling law cannot be fitted, we instead estimate the irreducible error as the minimum loss achieved: $E_{x|p} = \min_{N,D} L_x\left(f_p^{(N, D)}\right)$.
> > With $E_{x|p}, E_{y|p}$ from the compute-to-loss fits, we again use SciPy's `curve_fit` to fit $K, \kappa$ for the loss-to-loss scaling law  from Eq. (1).
>
> We add examples of compute-to-loss fits for some of the loss-to-loss scaling laws from Fig. 2 in this section; see new [Fig. 10](https://ibb.co/SqkwfwF). Note also that while Figs. 4-6 show averaged eval/validation performance, Figs. 22-27 in App. F includes results for specific validation and evaluation datasets (C4, The Pile UC, ARC-Easy, and HellaSwag).
>
> **[OSW Clarity 1a - More details on experimental setups]**: We have updated all the figure captions, Tab. 1, and [Tab. 2](https://ibb.co/xq3KgQ4N) in the appendix to include more details for the models used.
>
> **[OSW Clarity 1b - Clearer axes labels]**: Thank you for pointing this out. In our case, plot colors correspond to different dimensions in different plots. In Fig. 1, the x- and y-axis correspond to two specific datasets, and colors denote different interventions. In Fig. 2, the x-axis is fixed, and colors denote the dataset of the y-axis (most similar to Brandfonbrener). In Figs. 4-6, axes are again fixed (and y-axis reports an average over multiple datasets), and colors denote different values for the specific intervention. In all cases, the training set is specified in the caption. We have now updated the figure captions and legend layout to clarify this.
>
> **[W1 and Q2 - Modeling the influence of pretraining data]**: Quantitatively modeling the influence of data distributions is an intriguing open problem. The central issue is the difficulty in mapping differences in data distributions onto a simple feature space (e.g., scalar or vector values). In Data Mixing Laws [2], the mixing ratio serves this purpose effectively. However, we face the additional complexity of quantitatively comparing disjoint pretraining distributions with unknown compositions, a challenge beyond the scope of this work. We also do not believe multi-loss-to-loss scaling laws offer a satisfying solution. Without a reliable way to quantify diverse large-scale distributions and their relationships, we would need separate multivariate power laws for each combination of losses, again without a direct means for meaningful comparison.
>
> **[OSW2 and Q4 - Practical utility of the paper]**: We refer you to our response to Reviewer fuqS (section: **[W1 & Q2b - On the utility of loss-to-loss scaling curves]**) for an abridged version.
>
> We hope this addresses your concerns and encourages you to raise your score.

---

### Official Review · Reviewer_fuqS · 2025-03-20

**Overall Recommendation:** 2

**Summary:**

The paper investigates how loss-to-loss scaling (i.e. scaling laws between losses on different datasets) for LLMs is influenced by model architecture, tokenizer, and training datasets. The authors experimentally find that:

1. loss-to-loss scaling consistently follows shifted power laws.
2. The effects of pretraining data are more pronounced than the effects of model architecture and HPs.

## update after rebuttal"
I will keep the current score.

**Claims And Evidence:**

The claims are supported by evidence.

**Essential References Not Discussed:**

The paper discusses how the final loss depends upon pretraining data among other things. The recent paper "Scaling Optimal LR Across Token Horizons" also discusses how pretraining data will influence scaling properties.

**Experimental Designs Or Analyses:**

Experiments are sound

**Methods And Evaluation Criteria:**

See the strengths and Weaknesses section.

**Other Comments Or Suggestions:**

na

**Other Strengths And Weaknesses:**

Strengths:

1. The paper is very clearly written, and the claims it makes are well supported.
2. Scaling laws are impactful.
3. Identifying what interventions are effective for LLM training can save lots of compute.

Weaknesses.
1. The practical utility of the paper is limited. The authors demonstrated that training data has a larger effect on the scaling laws than model architecture. An alternative way to conclude that pretraining data is more important than architecture is to just use e.g. MMLU numbers. It is not clear how the methodology the authors propose would be better than this baseline approach to determine what interventions are effective and which are not.

**Questions For Authors:**

1. Could you discuss more in detail how the irreducible errors are estimated?
2. Could you somehow quantify how your approach is better at identifying what interventions are effective than a baseline approach which would just use e.g. MMLU numbers as a quality metric?

**Relation To Broader Scientific Literature:**

na

**Theoretical Claims:**

na

---

> ### Author Rebuttal · Authors · 2025-04-01
>
> We thank you for the helpful feedback. We address your concerns as follows:
>
> **[W1 & Q2a - On comparing raw numbers]**: Comparing individual performance metrics like MMLU is indeed possible and can illustrate the effectiveness of an intervention for a specific model scale and setting. For example, we added [Table 3](https://ibb.co/tpgMmP1Z) to the appendix, which shows the raw numbers for MMLU and several other tasks. Note that Figs. 22-27 in App. F already shows versions of Figs. 4-6 for specific validation and evaluation datasets (C4, The Pile UC, ARC-Easy, and HellaSwag).
> That said, comparisons on a single model scale and setting or on a single test set fall short in understanding whether the effectiveness of an intervention is dependent on model size, dataset size, downstream task or other factors. Additionally, evaluations on a single test set like MMLU are noisy, as evident from [Fig. 10](https://ibb.co/Rpn1xJRX), [Fig. 11](https://ibb.co/WNkR7SV3), [Fig. 12](https://ibb.co/jvPSBB2N), [Fig. 13](https://ibb.co/gZmwBTz0), [Fig. 14](https://ibb.co/sdJ3nLWz), and [Fig. 15](https://ibb.co/Zpn5c6k8), which we have added to the appendix. Our comprehensive study addresses this by systematically evaluating performance across multiple scales, factors, and downstream tasks while rigorously controlling all training parameters (including learning rate, optimizer, context length, tokenizer, model and dataset sizes).
>
> **[W1 & Q2b On the utility of loss-to-loss scaling curves]**: Loss-to-loss scaling laws across datasets provide critical insights beyond single-setting performance comparisons, as detailed in our related work and discussion sections. We reiterate the most important points here and have updated the corresponding sections in the paper to state this more clearly:
> - **Generalization**: Loss-to-loss scaling laws (train-to-train, train-to-test, or test-to-test) provide insight into how performance transfers across datasets (Taori et al., 2020, Fang et al., 2022, Brandfonbrener et al., 2024). Specifically, these scaling laws help answer the generalization question: if a model achieves a certain performance on a given dataset (generally training dataset), how well does it do on another task/dataset? As **Reviewer EE76** rightly points out, this can be of “broader interest to the community, in terms of advancing our understanding of the role of architecture, optimization, etc. in downstream properties.”
>
> - **Compute Budget Translation**: By combining train-to-test scaling laws with compute-to-train scaling laws, we can more precisely understand how compute budget translates into downstream task performance (Brandfonbrener et al., 2024), and also help uncover emergent model abilities (Du et al., 2025).
>
> - **Factor Decomposition**: Separately analyzing compute-to-train and train-to-downstream scaling laws helps isolate factors influencing each step. For instance, in our work, we identify that **architecture and optimizer settings do not influence loss-to-loss scaling laws — but they do affect compute-to-train scaling laws** (Kaplan et al., 2020; Hoffmann et al., 2022; Brandfonbrener et al., 2024; Porian et al., 2024; Li et al., 2025). As a result, architectures and optimizers can be independently optimized to enhance compute scaling without negatively impacting downstream task performance.
>
> - **Limitations of Single-Dataset Validation**: As **Reviewer EE76** highlights, our findings caution against relying solely on one validation dataset's loss (as typical in data curation methods, e.g., (Liu et al., 2024)) — achieving a particular loss on one validation set can correspond to varying downstream task performances. Thus, a single validation dataset may not comprehensively indicate downstream efficacy.
> By illuminating the factors that consistently impact loss-to-loss scaling laws, we enable practitioners and researchers to use them as a tool for analyzing and optimizing model training.
>
> **[Q1 - On the irreducible errors]**: We have added two Appendix sections that detail the scaling law formulation and explain how parameters are estimated; please refer to reply to **Reviewer EE76** for more details.
>
> **[Regarding the referenced paper]**: We are happy to include the suggested reference in our camera-ready version. As you rightly point out, the impact of pretraining data on loss-to-loss scaling laws has been shown before (Taori et al., 2020, Fang et al., 2022, Brandfonbrener et al., 2024), and our results further confirm this finding. However, to the best of our knowledge, our study is the first to show this comprehensively for a large number of datasets and model settings and is the **first to analyze the impact of other factors like architecture, optimizer settings, and context length**.
>
> We hope this addresses your concerns fully and encourages you to raise your score.
>
> (Li et al, 2025) (Mis)Fitting: A Survey of Scaling Laws

---

### Decision · Program_Chairs · 2025-05-01

**Decision:**

Accept (poster)

**Comment:**

The paper investigates factors that determine loss-to-loss scaling of LLMs. The paper find that the pretraining data and tokenizer determine the scaling trend, and model size, optimization hyperparameters, and model differences have less of an impact.

The paper received very mixed reviews (1,2,3,4):
- Reviewer fuqS (2) findes the practical utility limited. The authors emphasize that loss-to-loss scaling laws has several benefits: Understanding performance of one dataset as the function of performance on another; the finding that while model parameters (size, hyperparameters, architecture) impact compute scaling laws, they do not impact loss-to-loss scaling laws; and cautining against reyling on one validation sets loss `as typical in data curation methods’. I agree with the benefits of loss-to-loss scaling laws, but like to mention that it is not typical to rely on one validation set's loss in data curation methods.
- Reviewer EE76 (3) find the key finding that loss-to-loss scaling is not impacted by model parameters interesting. The reviewer notes, however, that the paper falls short in understanding how pre-traing data alters the loss-to-loss scaling.
- Reviewer he3E (4) find the claims generally well supported except that the models considered are relatively small, which is the reviewer's main concern. The authors clarified that they also based their results on larger models, and revised Fig 1 amongst other parts of the manuscript.
- Reviewer 6jji (1) raised concerns that were addressed by the reviewers.

I agree with reviewers EE76 and he3D that the paper's results are interesting and the claims are well supported. I therefore recommend acceptance. Reviewers EE76 and he3D also made several constructive comments that can help to further improve the data. An toy example or theoretical derivation on how loss-to-loss scaling laws can look like would also be interesting.